# The inhibition mechanism of the SUR2A-containing $K_{ATP}$ channel by a regulatory helix

Dian Ding[1,2,3,4,5], Tianyi Hou[1,2,3,4,5], Miao Wei[1,4], Jing-Xiang Wu[1,4] & Lei Chen [1,2,4] ✉

$K_{ATP}$ channels are metabolic sensors for intracellular ATP/ADP ratios, play essential roles in many physiological processes, and are implicated in a spectrum of pathological conditions. SUR2A-containing $K_{ATP}$ channels differ from other subtypes in their sensitivity to Mg-ADP activation. However, the underlying structural mechanism remains poorly understood. Here we present a series of cryo-EM structures of SUR2A in the presence of different combinations of Mg-nucleotides and the allosteric inhibitor repaglinide. These structures uncover regulatory helix (R helix) on the NBD1-TMD2 linker, which wedges between NBD1 and NBD2. R helix stabilizes SUR2A in the NBD-separated conformation to inhibit channel activation. The competitive binding of Mg-ADP with Mg-ATP to NBD2 mobilizes the R helix to relieve such inhibition, allowing channel activation. The structures of SUR2B in similar conditions suggest that the C-terminal 42 residues of SUR2B enhance the structural dynamics of NBD2 and facilitate the dissociation of the R helix and the binding of Mg-ADP to NBD2, promoting NBD dimerization and subsequent channel activation.

The opening of ATP-sensitive potassium channels ($K_{ATP}$) is inhibited by intracellular ATP but activated by Mg-ADP[1]. Therefore, $K_{ATP}$ channels convert cellular energy status (ATP/ADP ratio) into membrane potential and thus electrical signals[1]. Functional $K_{ATP}$ channels are localized on the plasma membrane and are composed of four Kir6 subunits and four SUR subunits[2]. Kir6 are inward-rectifier potassium channels that form the pore of $K_{ATP}$[2]. SUR subunits play essential regulatory roles and harbor the binding sites for drugs, including inhibitors (insulin secretagogues) and activators ($K_{ATP}$ openers)[3]. Mg-ADP activates the $K_{ATP}$ channels through SUR subunits as well[3]. There are three subtypes of SUR proteins in humans, SUR1, SUR2A, and SUR2B[4], which show broad tissue distribution[5]. SUR2A and SUR2B are two splicing variants that have the same length but differ in their C-terminal 42 residues (C42). The SUR1-containing $K_{ATP}$ channel in pancreatic islets is essential for insulin secretion. SUR2A plays important roles in the cardiac muscle and skeletal muscle, while SUR2B is distributed in the smooth muscle, such as blood vessels[2,5]. SUR2 proteins are encoded by the *ABCC9* gene. The loss-of-function mutations of the *ABCC9* gene cause dilated cardiomyopathy[6], familial atrial fibrillation[7], and intellectual disability myopathy syndrome[8], while the gain-of-function mutations of *ABCC9* lead to Cantu syndrome in humans[9,10]. Although SUR1, SUR2A, and SUR2B share high sequence homologies, they have differential responses to Mg-ADP activation and show variable sensitivities to drugs such as $K_{ATP}$ openers[11–15]. Mg-ADP activates SUR1 and SUR2B more strongly than SUR2A[11,12,16,17] and it was found that seven residues in the middle of C42 of SUR2A (amino acids 1516–1522) are critical for such a difference between SUR2A and SUR2B[18]. The differences in Mg-ADP activation of $K_{ATP}$ channels with different SUR

[1]State Key Laboratory of Membrane Biology, College of Future Technology, Institute of Molecular Medicine, Peking University, Beijing Key Laboratory of Cardiometabolic Molecular Medicine, 100871 Beijing, China. [2]Peking-Tsinghua Center for Life Sciences, Peking University, 100871 Beijing, China. [3]Academy for Advanced Interdisciplinary Studies, Peking University, 100871 Beijing, China. [4]National Biomedical Imaging Center, Peking University, 100871 Beijing, China. [5]These authors contributed equally: Dian Ding, Tianyi Hou. ✉e-mail: chenlei2016@pku.edu.cn

subtypes underlie their distinct physiological functions. SUR1-containing K$_{ATP}$ is the dedicated sensor of the intracellular Mg-ADP/Mg-ATP ratio in pancreatic cells to control hormone release in response to changes in blood glucose level, and it stays open under resting conditions[19]. In contrast, the SUR2A-containing K$_{ATP}$ channel in the heart keeps closed in normal conditions and only opens in response to severe metabolic inhibition, such as ischemic stress[20], ensuring that the K$_{ATP}$ channels in the heart do not disturb normal cardiac electrical function but could protect the heart from lethal injury[21].

SUR proteins belong to the type IV ABCC family of ABC transporters, which typically show a two-layer architecture: the transmembrane domain (TMD) layer and the cytosolic nucleotide-binding domain (NBD) layer[3]. The NBD layer is comprised of NBD1 and NBD2, which could dimerize to form two nucleotide-binding sites (NBS)[3]. In the SUR protein, only one NBS has conserved sequences for catalysis (the consensus site), thus the ATPase activity, while the other NBS (the degenerate site) has no ATPase activity due to the mutation in the key catalytic glutamate[3]. The helices of TMD1 and TMD2 are interweaved to form two structural halves that show large conformational changes during transport cycles. Two pairs of coupling helices bridge the TMD and NBD layers and mediate the bidirectional crosstalk between these two layers. The dimerization of NBDs drives the closure of two halves of TMD[22], while separated NBDs inhibit the closure of TMD. Structural studies on the pancreatic SUR1 protein reveal that in the presence of insulin secretagogues and ATP but without Mg, insulin secretagogues wedge themselves inside the central vestibule of TMD and stabilize SUR1 in the TMD-separated inward-facing conformation[23–29]. The ATP molecule binds in the degenerate site on NBD1 of SUR1, while no nucleotide was bound in the consensus site on NBD2. Therefore, we designate this structure as SUR1$_{Inward Facing/NBD1-ATP/NBD2-apo}$ (designated as SUR1$_{IF/ATP/apo}$). We use the same nomenclature for other structures throughout this paper unless indicated otherwise. Mg-ADP binding in the consensus site on NBD2 of SUR1 drives the asymmetric dimerization of NBD1 and NBD2, the closure of TMD, and the subsequent channel activation[26,30–33]. The recent structure of vascular SUR2B-Kir6.1 in the presence of ATP and the insulin secretagogue glibenclamide shows a similar inward-facing conformation of SUR2B (SUR2B$_{IF/ATP/apo}$)[34]. In addition, structures of SUR2A and SUR2B in complex with KCOs and Mg-ATP/Mg-ADP represent the activated NBD-dimerized occluded (OD) state (SUR2A$_{OD/MgATP/MgADP}$ and SUR2B$_{OD/MgATP/MgADP}$)[32]. Despite this progress, the detailed structural mechanism of how SUR2 proteins are regulated by Mg-nucleotides is not fully understood, especially why SUR2A responds differently to Mg-ADP activation compared to SUR1 or SUR2B is mysterious. Here we present the structures of SUR2A and SUR2B in the presence of different concentrations of Mg-ATP and Mg-ADP and an allosteric inhibitor, repaglinide (RPG). These structures highlight an inhibitory helix on SUR2A, and structural comparisons provide insights into the distinct regulatory mechanisms of SUR2A-containing K$_{ATP}$ channels in comparison to other K$_{ATP}$ channel isotypes.

## Results

### Structure of SUR2A in the presence of Mg-ATP and RPG

To reveal the mechanisms of SUR2 regulation by Mg-nucleotides, we sought to capture the intermediate states during activation by stabilizing SUR2 proteins in the TMD-separated conformation instead of the fully activated states that have been reported previously[32]. For this purpose, we used the non-selective insulin secretagogue repaglinide (RPG)[35] to block the complete conformational change of TMDs. To mimic two different nucleotide conditions inside the cells, we also supplemented 3 mM Mg-ATP for the high-ATP state or 1 mM Mg-ATP and 2 mM Mg-ADP for the high-ADP state into the protein sample for cryo-EM analysis. We have obtained the structures of SUR2 in these two different conditions (Supplementary Figs. 1–9 and Supplementary

Table 1). The SUR2A structure in the presence of Mg-ATP and RPG was determined at 2.8 Å resolution (Supplementary Fig.1 and Supplementary Table 1). SUR2A shows an inward-facing conformation that is overall similar to SUR1 in complex with RPG and ATP (SUR1$_{IF/ATP/Apo}$)[27,28] (Fig. 1a–e and Supplementary Fig. 1j). We observed the RPG density in the central vestibule of SUR2A (Figs. 1a, 1d, and 1f) and Mg-ATP densities in both NBD1 and NBD2 (Fig. 1g, h). Therefore, we designate this structure as SUR2A$_{IF/MgATP/MgATP}$. We did not observe the densities for the TMD0 domain (Fig. 1a, b, e), probably due to its high flexibility as observed previously in the structures of SUR2 in the presence of KCO and Mg-ATP/ADP[32].

### R helix wedges between NBD1 and NBD2 of SUR2

To our surprise, we observed a strong helical density that is sandwiched between NBD1 and NBD2 of SUR2A$_{IF/MgATP/MgATP}$ (Fig. 1a–c). Such density has never been observed in SUR1, which was expressed and purified similarly[36], ruling out the possibility that the density originates from common chromatographic contaminants. The density of this helix is connected to the C-terminus of NBD1 (Fig. 1e), and the connecting density could be observed better in the unsharpened map than in the sharpened map (Supplementary Fig. 2a, b), suggesting that this helix is part of the NBD1-TMD2 linker and that it connects to NBD1 through a region that has considerable structural instability. The excellent local map quality of the helix allowed us to resolve it as amino acids 924–942 on the NBD1-TMD2 linker (Supplementary Fig. 2c and Fig. 2a–d). To further validate whether the binding of this helix on SUR2A is energetically favorable, we isolated the structure of this helix and docked it onto the rest of the model of SUR2A using the HDOCK server[37] (Supplementary Fig. 2d–j). The HDOCK server successfully identified the binding pose of this helix on SUR2A with a reasonable docking score of −240.84 (Supplementary Fig. 2m), indicating an energetically favorable binding. To our knowledge, the presence of such a helix between NBD1 and NBD2 has neither been proposed nor observed in SUR proteins previously. Because the position of this helix is akin to the "R domain" of other ABCC transporters, such as CFTR[38], we designated it as the "Regulatory helix" (R helix). One side of the R helix interacts with NBD1. E929 forms a hydrogen bond with the main chain amino group of G811 and makes an electrostatic interaction with R815 of NBD1 (Fig. 2b). R930 makes an electrostatic interaction with D789, which also interacts with R815 of NBD1 (Fig. 2b). R930 forms an additional hydrogen bond with the main-chain carbonyl group of S785 in NBD1 (Fig. 2b). L933 and M937 bind inside a hydrophobic pocket surrounded by L792, L793, and I806 of NBD1 (Fig. 2b, c). The other side of the R helix interacts with NBD2 (Fig. 2b). R934 interacts with T1344 on the top of NBD2 (Fig. 2b). Y938 packs against ATP bound in NBD2, and its hydroxyl group makes a hydrogen bond with the γ-phosphate group of ATP (Fig. 2b, d). Both K931 and R935 on the R helix interact electrostatically with E1470 and ATP of NBD2 (Fig. 2d). The extensive interactions between the R helix and both NBDs stabilize SUR2A in the NBD-separated conformation (Fig. 1b–d). Sequence alignment shows that the R helix is conserved only in mammalian SUR2 but not in mammalian SUR1, in which six residues are inserted into the middle of the NBD1-TMD1 linker (between T932 and L933) and disrupt the precise spatial arrangement of NBD-interacting residues on the linker (Fig. 2e). Moreover, Y938 on the R helix of SUR2 is replaced by a Ser in SUR1 (Fig. 2e). These observations agree with the fact that no such helical structure has been observed previously in SUR1[29], neither in the SUR1-Kir6.2 octamer[23–28] nor in the SUR1 subunit alone[27,39].

### Mg-ADP binding on NBD2 mobilizes the R helix

In the cryo-EM sample of SUR2A with Mg-ATP/Mg-ADP and RPG, we observed two major 3D classes with the resolution of 3.2 Å and 3.8 Å, respectively (Supplementary Figs. 3 and 4 and Supplementary Table 1). The first 3D class is the same as SUR2A$_{IF/MgATP/MgATP}$ and both NBD1 and NBD2 are bound with Mg-ATP (Supplementary Fig. 3d, e). The

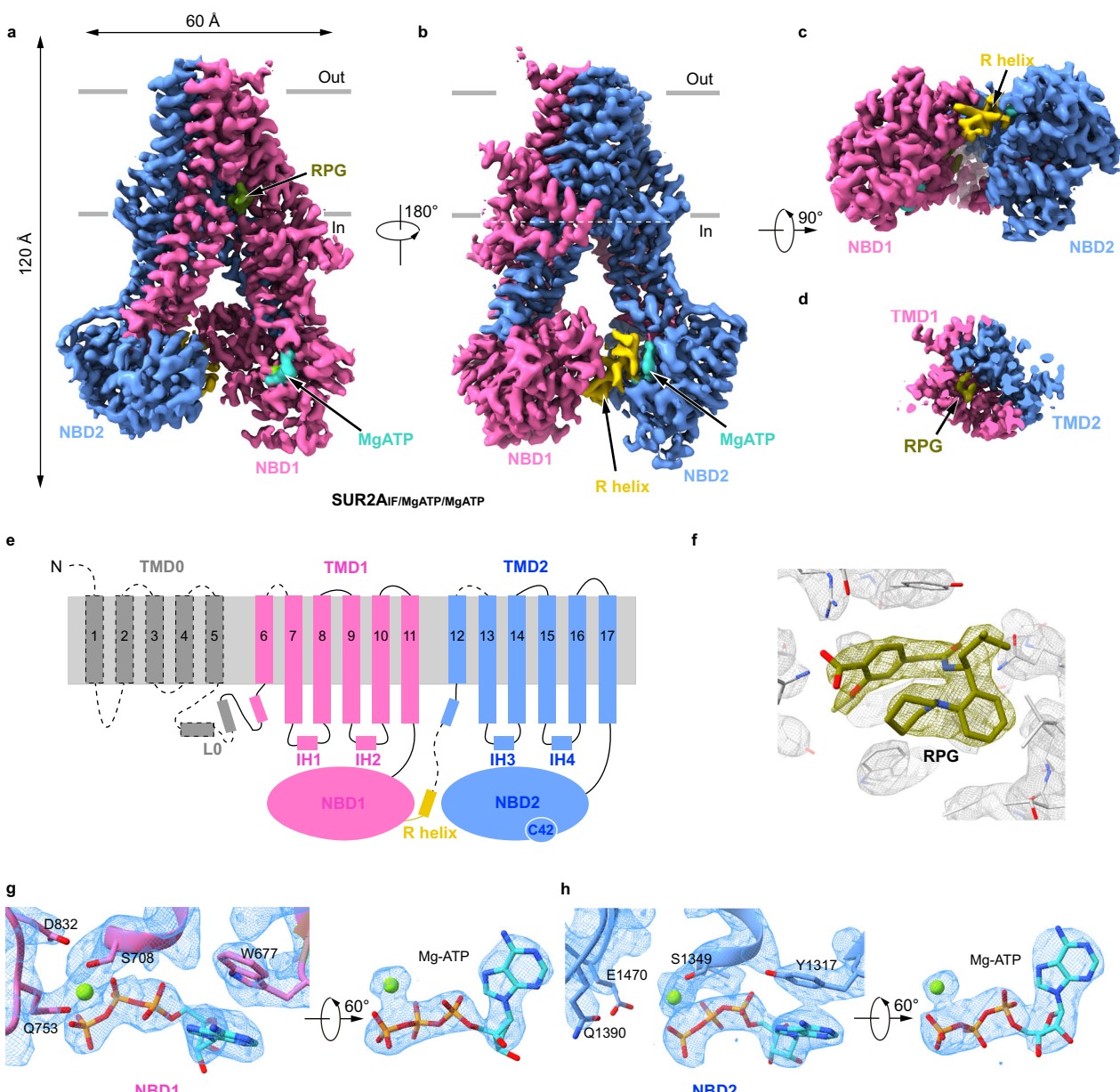

**Fig. 1 | Structure of SUR2A in complex with RPG and Mg-ATP. a–c** Cryo-EM density map of SUR2A$_{IF/MgATP/MgATP}$, viewed from the side (**a**, **b**) and bottom (**c**). The approximate position of the lipid bilayer is indicated by gray bars. Transmembrane domain1 (TMD1)- nucleotide binding domain1 (NBD1), TMD2-NBD2, R helix, Repaglinide (RPG), Mg$^{2+}$, and ATP are colored in pink, blue, olive, green, and cyan, respectively. To better visualize the density of the RPG, part of the transmembrane helix density in the front of the RPG was omitted. **d** The cut-open view of the transmembrane domain at the position of the cross-section indicated by the white dashes in (**b**). **e** Topology of SUR2 subunit in cartoons. **f** Electron density of RPG in SUR2A$_{IF/MgATP/MgATP}$. The map is shown as mesh. Protein and RPG are shown as sticks. The map was contoured at 1.01 level (7.3 σ). **g**, **h** Close-up views of the EM densities of nucleotides bound in NBD1 (**g**) and NBD2 (**h**). The maps were contoured at 0.96 level (6.9 σ) (**g**) and 0.74 level (5.4 σ) (**h**), respectively.

second 3D class also shows an inward-facing conformation, which is overall similar to SUR2A$_{IF/MgATP/MgATP}$ with an RMSD of 1.698 Å. In the density map of the second 3D class, we found that RPG is bound in TMD and there are nucleotide densities on both NBDs (Supplementary Fig. 4g–i). However, two obvious features of the second 3D class suggest that it is in a different conformation compared to the first 3D class: first, there is no R helix density between the two NBDs in the second 3D class (Fig. 3a, b); second, the two NBDs of the second 3D class move closer to each other, and the RMSD at the NBD layer is 2.204 Å, which is markedly larger than the overall RMSD (Fig. 3c–f). Such difference can also be readily observed in the superposed electron density maps, both of which were low-pass filtered to 6 Å for fair comparison

(Supplementary Fig. 4k–n). Unfortunately, the local map quality of NBDs was not sufficient for the assignment of the nucleotide identities purely based on the electron density map (Supplementary Fig. 4g, h). Because NBD1 was always bound with Mg-ATP even in the presence of a high concentration of Mg-ADP[32], we assigned the nucleotide bound at NBD1 as Mg-ATP. Moreover, because of the different conformation of the second 3D class compared with the first 3D class (SUR2A$_{IF/MgATP/MgATP}$), we reasoned that Mg-ADP rather than Mg-ATP might bind at NBD2, otherwise the second 3D class would have the same structure as SUR2A$_{IF/MgATP/MgATP}$. Notably, although the assignment of Mg-ADP at NBD2 is logically reasonable and the structural models, especially the coordinates of nucleotides, were refined against

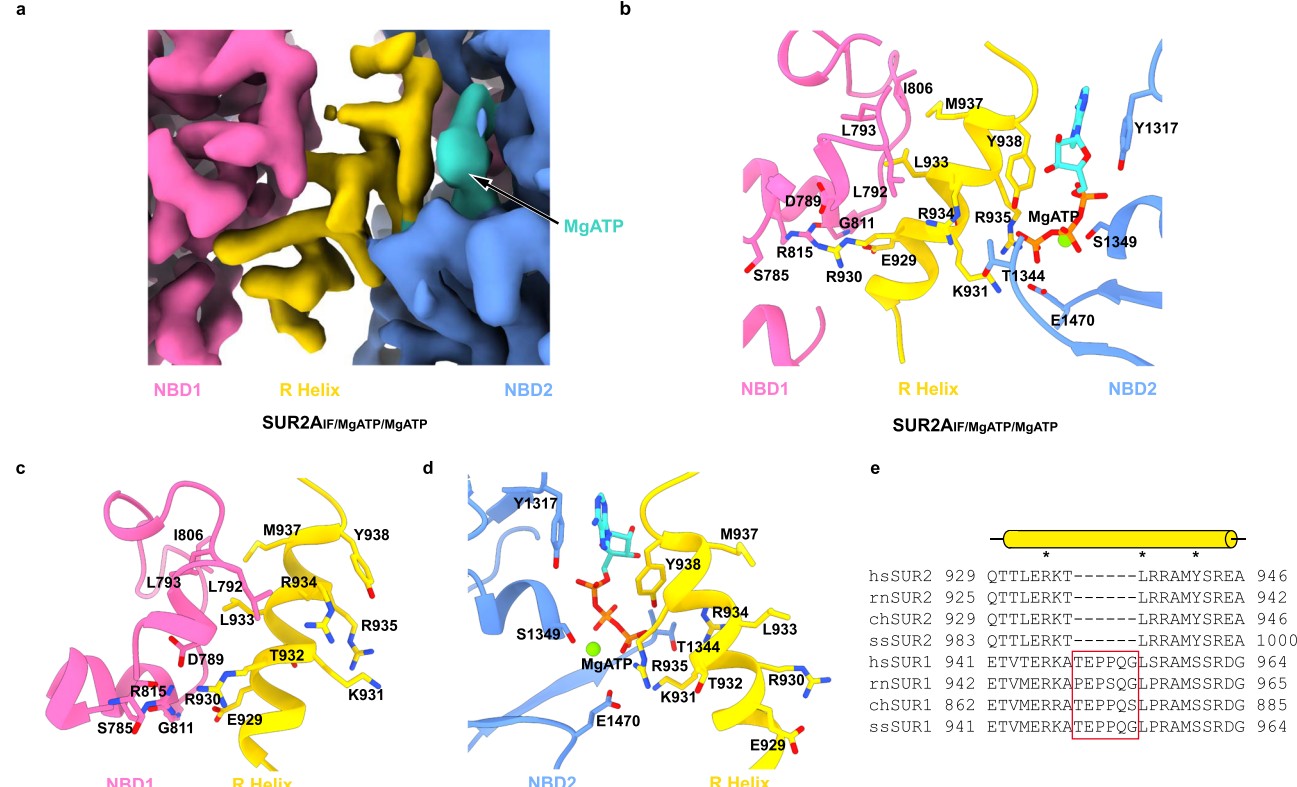

**Fig. 2 | The regulatory helix binds to both NBD1 and NBD2 of SUR2A. a** The Close-up view of the electron density map at the R helix region. The color scheme is the same as that in Fig. 1. The map was contoured at 0.63 level (4.6 σ). **b** The binding between the R helix and the two NBDs of SUR2A$_{IF/MgATP/MgATP}$ in the same view as in (**a**). The amino acids involved in the interactions between the R helix and NBD are shown as sticks. **c** The detailed interactions between the R helix and NBD1. **d** The detailed interactions between the R helix and NBD2. **e** Sequence alignment of the R helix from *Homo sapiens* SUR1 (hsSUR1), *Rattus norvegicus* SUR1 (rnSUR1), *Capra hircus* SUR1 (chSUR1), *Sus scrofa* SUR1 (ssSUR1), and SUR2 from the same species. R helix is shown as a yellow cylinder. The amino acids on the R helix that are important for interaction with NBD1 and NBD2 are marked with asterisks above. The insertions in SUR1 are boxed in red.

the cryo-EM maps to reasonable geometry, we suggest cautious interpretation of the nucleotide-binding poses, because of the large positional uncertainty intrinsic to this local map quality (Supplementary Fig. 4g, h). Based on the discussion aforementioned, we tentatively assign the second 3D class as SUR2A$_{IF/MgATP/MgADP}$. Compared to SUR2A$_{IF/MgATP/MgATP}$, the two NBDs of SUR2A$_{IF/MgATP/MgADP}$ move closer to each other (Fig. 3d–f) and the binding site of the R helix is disrupted due to steric clashes between the R helix and the two NBDs of SUR2A$_{IF/MgATP/MgADP}$ (Fig. 3g, h). We also tried to dock the R helix onto the structure of SUR2A$_{IF/MgATP/MgADP}$ using the HDOCK server but the server could not find a reasonable binding site for the R helix. This docking analysis correlates with our structural observation that the R helix is not observed in the cryo-EM density map of SUR2A$_{IF/MgATP/MgADP}$ (Fig. 3a, b).

### Structural dynamics of SUR2B NBD2

Because SUR2B responds differently to Mg-nucleotides compared to SUR2A, we sought to understand the underlying structural basis. The single-particle analysis of SUR2B in the presence of Mg-ATP and RPG revealed two prominent 3D classes (Supplementary Fig. 5 and Supplementary Table 1). One 3D class has some residual helical densities of TMD but no density of NBD2 and could not be refined to high resolution (Supplementary Fig. 5c). It is possible that this class represents the damaged particles on the grids. However, given the fact that the cryo-EM sample preparation conditions of SUR2A and SUR2B were the same and the 3D class without NBD2 was not observed in the sample of SUR2A, the missing NBD2 is highly likely due to the large mobility of SUR2B NBD2, similar to that observed in the Δ508 CFTR structure, where NBD1 is missing in the map without correctors[40]. The other 3D

class shows prominent NBD2 density, and the reconstruction reached 3.7 Å resolution (Supplementary Figs. 5 and 6 and Supplementary Table 1). This structure resembles SUR2A$_{IF/MgATP/MgATP}$ and shows the density of the R helix and Mg-ATP (Fig. 4a, b). Therefore, we designate it as SUR2B$_{IF/MgATP/MgATP}$. However, the densities of NBDs are blurry compared to that of TMD, indicating the higher structural dynamics of NBDs (Supplementary Fig. 5f, g).

The image analysis of the SUR2B sample in the presence of RPG and Mg-ATP/Mg-ADP also reveals two dominant 3D classes, both of which reach 3.6 Å resolution (Supplementary Figs. 7–9 and Supplementary Table 1). The first 3D class is similar to the SUR2A$_{IF/MgATP/MgADP}$ structure, with Mg-ATP bound in NBD1 and Mg-ADP bound in NBD2, and with weak and discontinuous R helix density (Fig. 4c, d and Supplementary Fig. 8). We designate it as SUR2B$_{IF/MgATP/MgADP}$. The second 3D class has Mg-ATP bound in NBD1, Mg-ADP bound in NBD2, and RPG bound in the TMD (Supplementary Fig. 9e–g), but no R helix density was observed (Fig. 5a–c). Moreover, NBD1 and NBD2 move even closer to each other than SUR2B$_{IF/MgATP/MgADP}$, but both the consensus site and the degenerate site are not fully closed in comparison to SUR2B$_{OD/MgATP/MgADP}$ (Fig. 5d–l). We measured the distances of two NBDs using marker atoms (between G811 and S1346 and between G1448 and C707) and found obvious differences (Fig. 5j–l). The movement of NBDs correlates with the conformational change of TMD, in which two halves of TMD approach inwardly towards each other (Fig. 5d–f). However, the binding of RPG precludes the full closure of TMD (Fig. 5g–i). As a result, both the spatial arrangement of NBD1 and NBD2 (Fig. 5j–l) and the conformation of TMD (Fig. 5f, i) are in the middle of SUR2B$_{IF/MgATP/MgADP}$ and SUR2B$_{OD/MgATP/MgADP}$, suggesting it

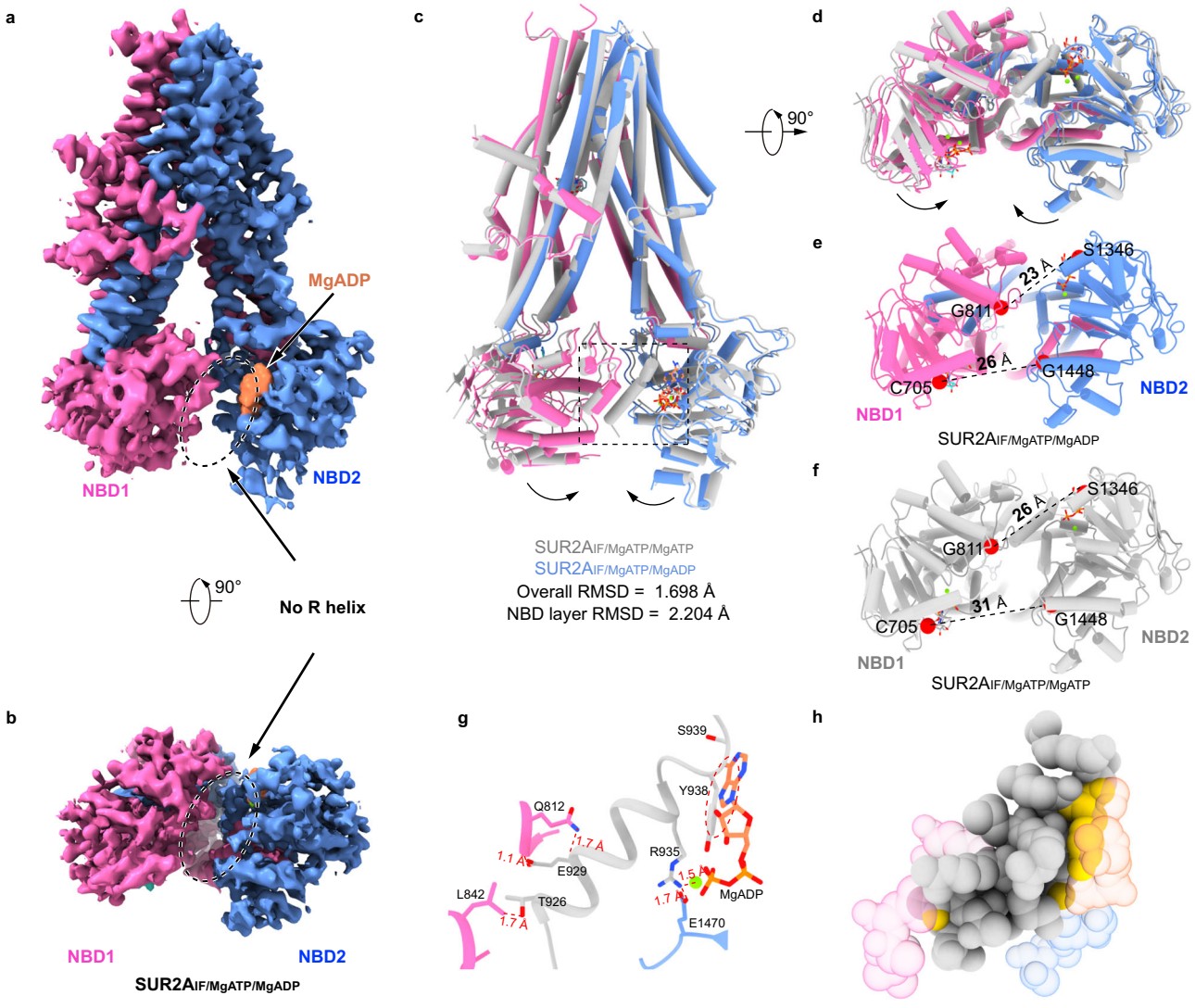

**Fig. 3 | Structure of SUR2A in complex with RPG, Mg-ATP, and Mg-ADP.**
**a**, **b** Cryo-EM density map of SUR2A$_{IF/MgATP/MgADP}$ viewed from the side (**a**) and bottom (**b**). ADP is colored in orange and the color scheme of the remaining parts is the same as that in Fig. 1. The empty R helix binding site is denoted with a dashed oval. **c**, **d** Structural alignment of SUR2A$_{IF/MgATP/MgATP}$ (gray) and SUR2A$_{IF/MgATP/MgADP}$ (colored) viewed from the side (**c**) and bottom (**d**). Arrows denote the movements of NBDs from SUR2A$_{IF/MgATP/MgATP}$ to SUR2A$_{IF/MgATP/MgADP}$. **e**, **f** The bottom views of the NBD layer of SUR2A in different states. The Cα distances between glycine in the Walker A motif and serine (cysteine at the degenerate site) in the ABC signature motif (G811-S1346 and G1448-C705) of SUR2A$_{IF/MgATP/MgADP}$ (**e**) and SUR2A$_{IF/MgATP/MgATP}$ (**f**) are shown as dashed lines. Cα atoms are shown as red spheres. **g** Sterical clashes between the R helix of SUR2A$_{IF/MgATP/MgATP}$ (gray) and both NBDs of SUR2A$_{IF/MgATP/MgADP}$ (colored) are boxed as dashes in (**c**). Distances between atoms that are shorter than 2.2 Å were shown. The red dashed oval indicates several sterical clashes between Y938 of the R helix and ATP bound on NBD2. **h** The atoms in (**g**) were shown as spheres. Atoms on the R helix (gray) that clash with both NBDs of SUR2A$_{IF/MgATP/MgADP}$ are colored in yellow, suggesting that the R helix is not compatible with binding to SUR2A$_{IF/MgATP/MgADP}$.

represents an intermediate state of SUR2B during activation, which is in transition from the inward-facing state to the occluded state. Therefore, we designate this structure as the partially occluded (PO) state (SUR2B $_{PO/MgATP/MgADP}$).

## R helix inhibits the dimerization of SUR2 NBDs

Since the R helix wedges between NBD1 and NBD2, it might regulate NBD dimerization and the subsequent activation of SUR2. To understand the regulatory function of the R helix, we mutated residues on the R helix that interact with NBD1 or NBD2 in SUR2 into alanines (Fig. 6a) and measured the currents of the SUR2-Kir6.2 K$_{ATP}$ channel in the presence of different Mg-nucleotides (Fig. 6b and Supplementary Fig. 10). We found that mutations of residues that interact with NBD1 (R930A and L933A) or NBD2 (Y938A), enhance the SUR2A activation by Mg-ATP or Mg-ADP (Fig. 6b), indicating that the R helix inhibits the activation of the SUR2A-containing K$_{ATP}$ channel and that R930, L933,

and Y938 play key roles in the binding of the R helix to the NBDs of SUR2A. Moreover, a 6-residue insertion, which mimics the corresponding sequence of SUR1, also significantly enhances the activation of SUR2A by Mg-ATP/ADP (Fig. 6b), in agreement with the fact that the NBD1-TMD2 linker of SUR1 does not inhibit the K$_{ATP}$ channel in the same manner and no R helix structure has been observed in SUR1[23–29].

Gain-of-function mutations of SUR2 cause Cantu syndrome[9,10]. The D793V mutation in human SUR2 (D789V in rat SUR2) was identified in patients with Cantu syndrome and was shown to cause hyperactivation of SUR2-containing K$_{ATP}$ channel[41]. However, D789 and its adjacent residues on NBD1 of SUR2 do not form obvious interactions with residues on NBD2 when NBDs are dimerized in the structure of SUR2A$_{OD/MgATP/MgADP}$ (PDB ID: 7VLU)[32] (Supplementary Fig. 11k), indicating that D789V might not affect the stability of NBD dimer. In SUR2A$_{IF/MgATP/MgATP}$, D789 on NBD1 interacts with R930 on the R helix (Fig. 2c), and mutation of D789V would disrupt such an interaction to

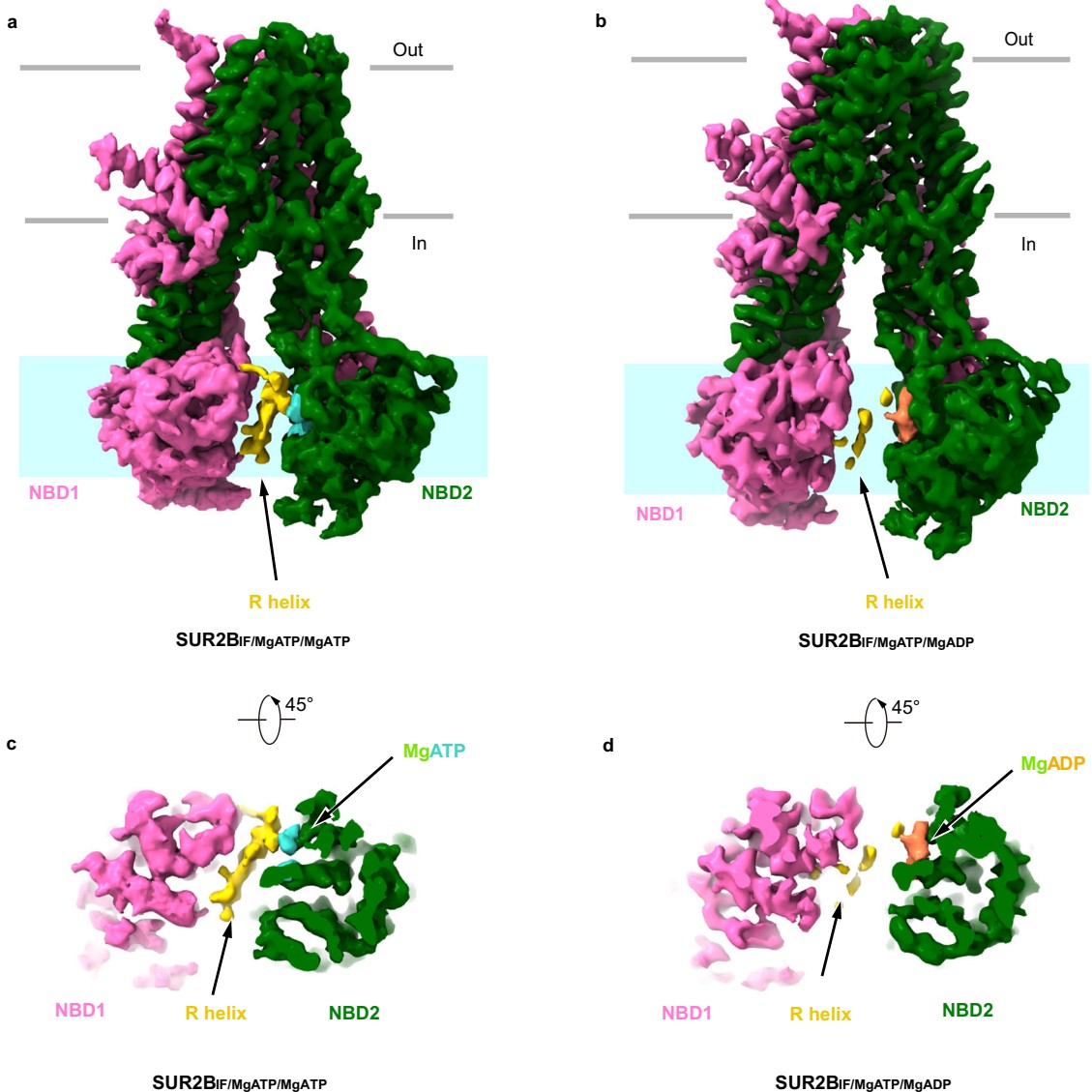

**Fig. 4 | Structure of SUR2B$_{IF/MgATP/MgATP}$ and SUR2B$_{IF/MgATP/MgADP}$. a** The cryo-EM density map of SUR2B$_{IF/MgATP/MgATP}$, viewed from the side. **b** The cryo-EM density map of SUR2B$_{IF/MgATP/MgADP}$, viewed from the side. **c** The cryo-EM density map of SUR2B$_{IF/MgATP/MgATP}$, viewed from a 45° rotated view. **d** The cryo-EM density map of SUR2B$_{IF/MgATP/MgADP}$, viewed from a 45° rotated view. Nucleotide-binding domain1 (NBD1), NBD2, and the R helix are colored in pink, dark green, and yellow, respectively. The approximate position of the layer shown in (**c**, **d**) is denoted by cyan rectangles in (**a**, **b**), respectively.

relieve the inhibition of the R helix, leading to the hyperactivation of SUR2A by Mg-ATP/ADP. In agreement with this, an additional mutation of D789-interacting R930 on the R helix (the R930A, D789V double mutant) does not further activate the D789V mutant (Fig. 6b). In contrast to the significant activation of the SUR2A-containing K$_{ATP}$ channel by R helix mutations, the activation of the SUR2B-containing channel by such mutations, such as R930A and Y938A, is marginal and not statistically significant (Fig. 6b).

## Discussion

Our structural and functional studies reveal that the R helix inhibits NBD dimerization and thus SUR2A activation (Figs. 6 and 7). The built-in regulatory elements in ABC transporters were previously identified in several other ABCC family members. For example, the well-studied R domains of both CFTR (ABCC7)[42] and yeast Ycf1p (an yeast ABCC transporter)[43,44] are located on the NBD1-TMD2 linker and negatively regulate the activities of these proteins. The inhibition of CFTR and Ycf1p by the R domain is relieved upon phosphorylation. Although it is unknown whether the R helix of SUR2 is subjected to phosphorylation

as well, there are several modifiable residues on the R helix, such as tyrosine and lysine (Fig. 2e), suggesting the possibility of post-translational modifications, such as phosphorylation and methylation. In addition, our study suggests the gain-of-function D789V mutation found in patients with Cantu syndrome possibly activates the SUR2A-containing channel by releasing the R helix (Fig. 6b). Moreover, the non-conserved sequence of NBD1-TMD1 linker in SUR1 (Fig. 2e) correlates with the fact that no R helix is observed in SUR1 and underlies the fundamental difference between SUR1 and SUR2A in Mg-nucleotide activation: SUR1-containing K$_{ATP}$ channels could be activated much easier than SUR2A-containing K$_{ATP}$ channels. Notably, the R helix was not observed in the structure of the SUR2B-Kir6.1 complex reported recently[34], possibly because of the low local map quality of NBDs or because no Mg was supplemented into the protein sample in that study.

SUR2B is activated by Mg-ATP easier than SUR2A (Fig. 6b). This agrees with the stable structure of SUR2A$_{IF/MgATP/MgATP}$ NBD layer (Supplementary Fig. 1f–i) and the dynamic structure of SUR2B$_{IF/MgATP/MgATP}$ NBD layer (Supplementary Fig. 5f, g) manifested by the local

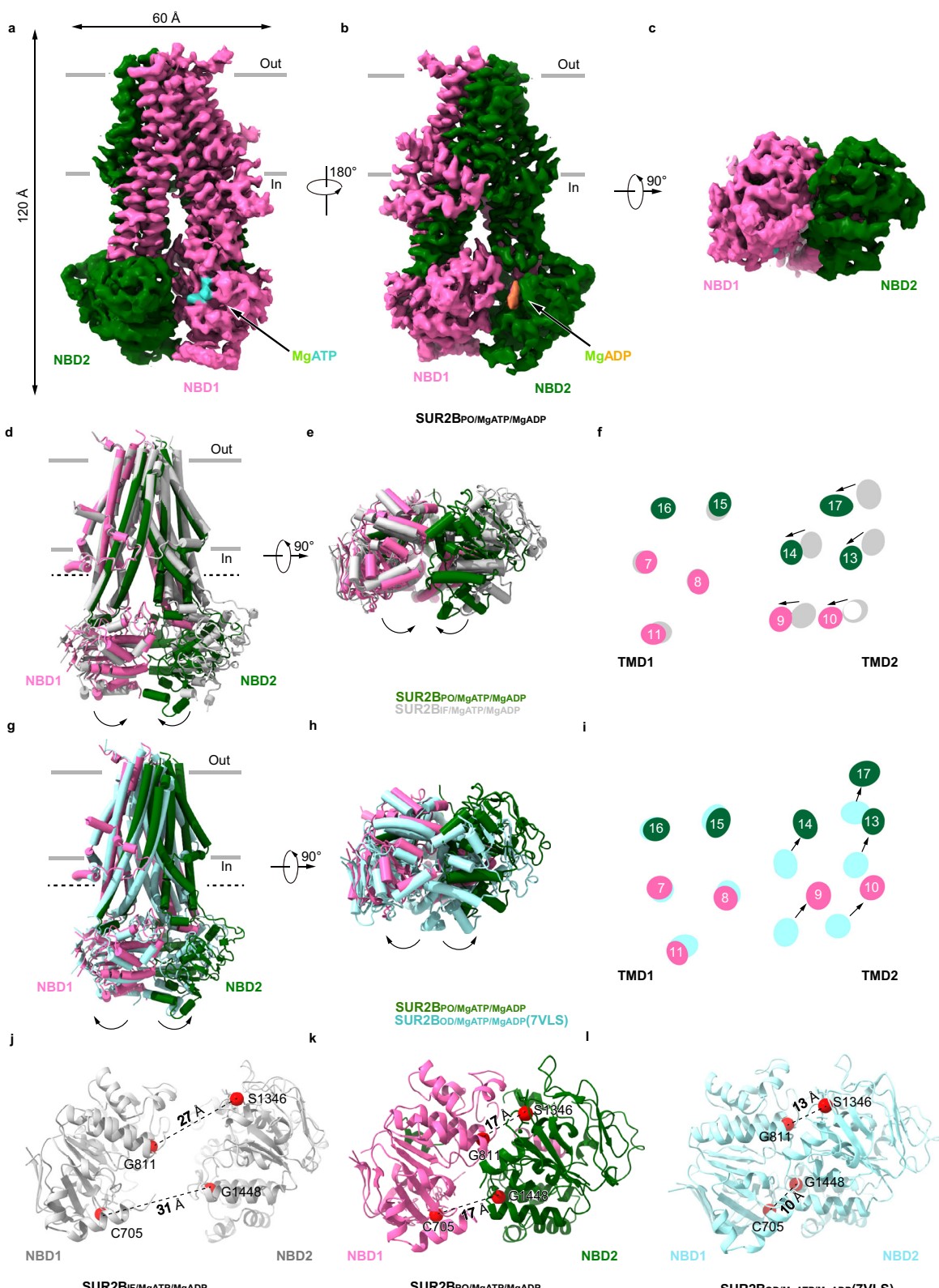

resolution map and B factor distribution in the same condition as Mg-ATP and RPG. In addition, one class of SUR2B in the presence of Mg-ATP even shows no density of NBD2, suggesting its high flexibility (Supplementary Fig. 5). Because SUR2B and SUR2A only differ in their C42, which do not interact with the R helix or NBD1, SUR2B-C42 might somehow enhance the dynamics of NBD2 to destabilize the R helix-bound, NBD-separated inactive conformation.

In addition, SUR2B shows higher activation by Mg-ADP than SUR2A (Fig. 6b), which also correlates well with our structural observations. In these structures, Mg-ATP is constantly bound in NBD1, while Mg-ADP or Mg-ATP could competitively bind in NBD2. Moreover, in the presence of 2 mM Mg-ATP and 1 mM Mg-ADP, 70% of the resolvable SUR2A (143,204 particles) has Mg-ATP bound in NBD2, and the remaining 30% of SUR2A (61,007 particles) has Mg-ADP bound in

**Fig. 5 | Structure of SUR2B$_{PO/MgATP/MgADP}$. a–c** The cryo-EM density map of SUR2B$_{PO/MgATP/MgADP}$, viewed from the side (**a, b**) and bottom (**c**). The approximate position of the lipid bilayer is indicated by gray bars. TMD1-NBD1, TMD2-NBD2, RPG, Mg$^{2+}$, ADP, and ATP are colored in pink, dark green, olive, green, coral, and cyan, respectively. **d–f** Structural comparison between SUR2B$_{IF/MgATP/MgADP}$ state and SUR2B$_{PO/MgATP/MgADP}$ state, in the side view (**d**), the bottom view (**e**), and the central section (**f**). Structures are aligned according to their TMD1 domains. The approximate position of the cross-section shown in (**f**) is indicated by the dashed line in (**d**). TMD1-NBD1 and TMD2-NBD2 of SUR2B$_{PO/MgATP/MgADP}$ are colored in pink and dark green, respectively. SUR2B$_{IF/MgATP/MgADP}$ is colored in gray. Arrows denote the movements from SUR2B$_{IF/MgATP/MgADP}$ to SUR2B$_{PO/MgATP/MgADP}$. **g–i** Structural comparison between SUR2B$_{OD/MgATP/MgADP}$ state and SUR2B$_{PO/MgATP/MgADP}$ state, in

the side view (**g**), the bottom view (**h**), and the central section (**i**). Structures are aligned according to their TMD1 domains. The approximate position of the cross-section shown in (**i**) is indicated by the dashed line in (**g**). TMD1-NBD1 and TMD2-NBD2 of SUR2B$_{PO/MgATP/MgADP}$ are colored in pink and dark green, respectively. SUR2B$_{OD/MgATP/MgADP}$ is colored in cyan. Arrows denote the movements from SUR2B$_{OD/MgATP/MgADP}$ to SUR2B$_{PO/MgATP/MgADP}$. **j–l** Bottom view of the NBD layer of SUR2B in different states. The Cα distances between glycine in the Walker A motif and serine (cysteine at the degenerate site) in the ABC signature motif (G811-S1346 and G1448-C705) of SUR2B$_{IF/MgATP/MgADP}$ (**j**), SUR2B$_{PO/MgATP/MgADP}$ (**k**), and SUR2B$_{OD/MgATP/MgADP}$ (**l**) are shown as dashed lines. NBD1 is colored in pink, and NBD2 is colored in dark green. Cα atoms are shown as red spheres.

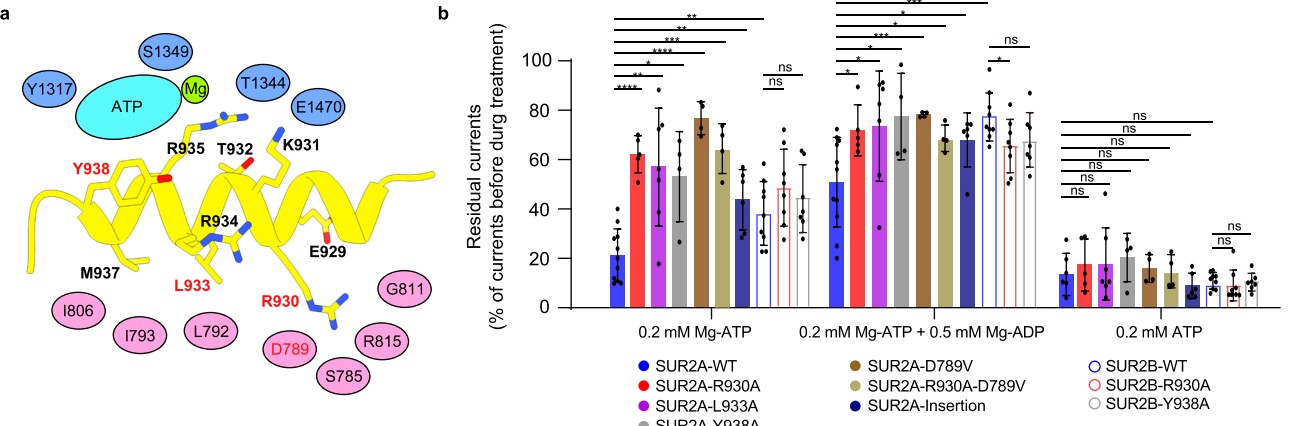

**Fig. 6 | R helix inhibits the activation of the SUR2A-containing K$_{ATP}$ channel. a** A cartoon model of the interactions between the R helix (yellow) and NBD1 (pink) and NBD2 (blue) of SUR2A. Interacting residues on the R helix are shown as sticks. Interacting residues on NBDs are shown as ovals. Mutated residues in (**b**) are labeled in red. **b** The effects of Mg-ATP, Mg-ATP/Mg-ADP, and ATP on various K$_{ATP}$ constructs, measured by inside-out patch. Currents after drug treatment were normalized to currents before drug application. Data are presented as mean values ± SD. The numbers of independent experiments of SUR2A-WT, SUR2A-R930A, SUR2A-I933A, SUR2A-Y938A, SUR2A-D789V, SUR2A-R930A-D789V, SUR2A-Insertion, SUR2B-WT, SUR2B-R930A, and SUR2B-Y938A are 11, 5, 7, 4, 4, 6, 9, 8 and 7, respectively. *P* value of effects of Mg-ATP between SUR2A-WT and SUR2A-R930A, SUR2A-I933A, SUR2A-Y938A, SUR2A-D789V, SUR2A-R930A-D789V, SUR2A-Insertion, SUR2B-WT are $2.7 \times 10^{-6}$, $6.5 \times 10^{-3}$, $3.3 \times 10^{-2}$, $8.8 \times 10^{-7}$, $4.8 \times 10^{-4}$,

$4.1 \times 10^{-3}$ and $8.4 \times 10^{-3}$, respectively. *P* value of effects of Mg-ATP between SUR2B-WT and SUR2B-R930A, SUR2B-Y938A are 0.16 and 0.39, respectively. *P* value of effects of Mg-ATP/Mg-ADP between SUR2A-WT and SUR2A-R930A, SUR2A-I933A, SUR2A-Y938A, SUR2A-D789V, SUR2A-R930A-D789V, SUR2A-Insertion, SUR2B-WT are $1.2 \times 10^{-2}$, $4.6 \times 10^{-2}$, $4.5 \times 10^{-2}$, $5.4 \times 10^{-4}$, $1.2 \times 10^{-2}$, $2.9 \times 10^{-2}$, and $8.5 \times 10^{-4}$, respectively. *P* value of effects of Mg-ATP/Mg-ADP between SUR2B-WT and SUR2B-R930A, SUR2B-Y938A are $3.3 \times 10^{-2}$ and 0.10, respectively. *P* value of effects of ATP between SUR2A-WT and SUR2A-R930A, SUR2A-I933A, SUR2A-Y938A, SUR2A-D789V, SUR2A-R930A-D789V, SUR2A-Insertion, SUR2B-WT are 0.50, 0.51, 0.28, 0.55, 0.77, 0.19, and 0.08, respectively. *P* value of effects of ATP between SUR2B-WT and SUR2B-R930A, SUR2B-Y938A are 0.48 and 0.78, respectively (*$P < 0.05$, **$P < 0.01$, ***$P < 0.001$, ****$P < 0.0001$ by two-side *t* test).

NBD2 (Supplementary Fig. 3c). In contrast, most of SUR2B has Mg-ADP bound in NBD2 (Supplementary Figs. 7c, 8 and 9). The differential occupancies of Mg-ADP in NBD2 suggest that SUR2B binds to Mg-ADP more tightly than SUR2A under the same condition with 2 mM Mg-ATP and 1 mM Mg-ADP. Furthermore, when NBD2 is bound with Mg-ADP, the R helix is either highly mobile in SUR2B$_{IF/MgATP/MgADP}$ (Fig. 4c, d) or completely dissociated in SUR2A$_{IF/MgATP/MgADP}$ (Fig. 3a) and SUR2B$_{PO/MgATP/MgADP}$ (Fig. 5c), suggesting the occupancy of Mg-ADP in NBD2 negatively correlates with the structural stability of the inhibitory R helix (Fig. 7b, c). This is probably because the γ-phosphate of ATP bound in NBD2 plays a key role in interacting with Y938 on the R helix (Figs. 2d and 6b) and the displacement of ATP by ADP disrupts such interaction, leading to the dissociation of the R helix. Since C42 does not directly interact with Mg-nucleotides (Fig. 2), SUR2B-C42 likely allosterically promotes the binding of Mg-ADP to NBD2.

The excellent local map quality of SUR2A$_{IF/MgATP/MgATP}$ allows the unambiguous model building of the C42 region in this conformation (Supplementary Fig. 2a, b). We found that the C42 of SUR2A$_{IF/MgATP/MgATP}$ is highly similar to that of SUR2A$_{OD/MgATP/MgADP}$ or SUR2B$_{OD/MgATP/MgADP}$ reported previously, with RMSDs of 1.631 Å and 1.724 Å[32], respectively (Supplementary Fig. 11a, b). Although the densities of C42 in other maps are blurry because of the dynamics of the

associated NBD2 (Supplementary Figs. 4j, 6f, g, 8h, i, and 9h, i) and do not allow the accurate modeling of side chains, the continuous electron densities support our tracing of the C42 main chains, resulting in similar structures. It is likely that although C42 of SUR2A and SUR2B share a common structure, their distinct amino acid compositions might differentially affect the conformational stability and Mg-nucleotide binding of NBD2 through modulation of its thermodynamics.

Structural comparison of SUR2A between inward-facing (SUR2A$_{IF/MgATP/MgATP}$ and SUR2A$_{IF/MgATP/MgADP}$) and occluded conformations (SUR2A$_{OD/MgATP/MgADP}$)[32] (Supplementary Fig. 11c–f) reveals that NBD2 contracts upon Mg-ADP binding and NBD dimerization (Supplementary Fig. 11d). More interestingly, a similar analysis shows the NBD2 of SUR2B contracts gradually upon Mg-ADP binding (Supplementary Fig. 11h, i, k, l). The degree of NBD2 contraction is SUR2B$_{IF/MgATP/MgATP}$ < SUR2B$_{PO/MgATP/MgADP}$ < SUR2B$_{OD/MgATP/MgADP}$ (Supplementary Fig. 11h, i, k, l), suggesting that Mg-ADP-induced contraction of SUR2B NBD2 precedes the dimerization of NBDs. In contrast, the NBD1 constantly binds Mg-ATP, and its conformation stays the same, irrespective of different SUR2 isotypes or functional states (Supplementary Fig. 11c, e, g, j). These observations further support the hypothesis that NBD2 is the allosteric sensor for Mg-ADP in SUR2 (Fig. 7a–c)[32], akin to SUR1[26,30].

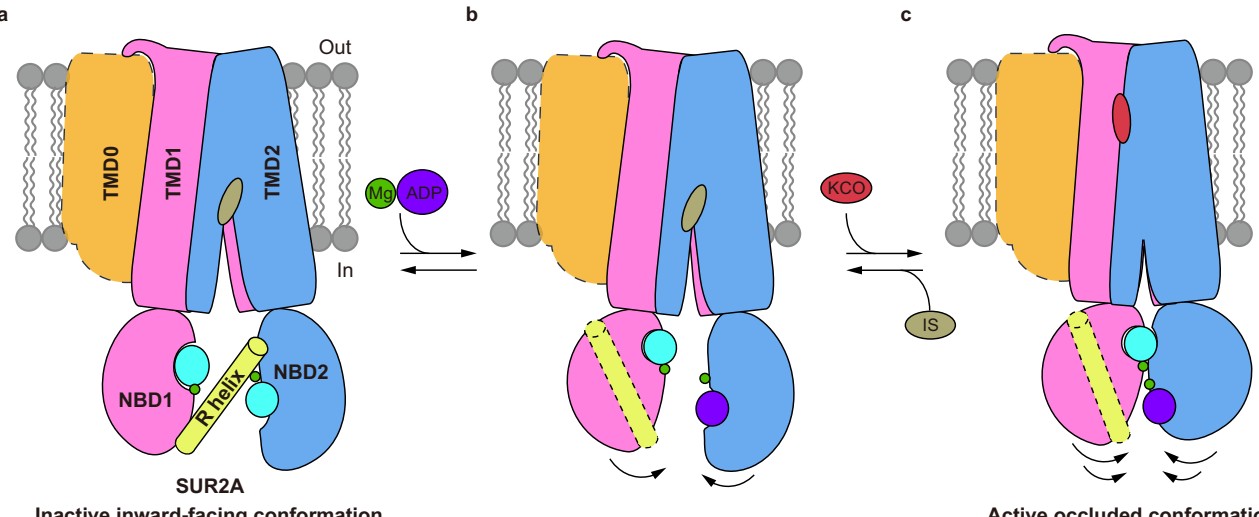

**Fig. 7 | Regulation of SUR2A subunit activation by the R helix. a–c** Side view of the cartoon model of SUR2 subunits. TMD0, TMD1-NBD1, TMD2-NBD2, the R helix, $Mg^{2+}$, ATP, ADP, insulin secretagogue (IS), and $K_{ATP}$ opener (KCO) are colored in orange, pink, blue, yellow, green, cyan, purple, gray, and red, respectively. TMD0 is outlined with dashes due to its flexibility in cryo-EM maps. In the presence of Mg-ATP, the R helix is stably bound between NBD1 and NBD2, inhibiting their closure (**a**). When Mg-ADP concentration is increased, Mg-ADP competes with Mg-ATP to bind to NBD2 to mobilize the R helix (**b**). When the R helix dissociates from two NBDs, NBD1 and NBD2 dimerize, which in turn activate the Kir6 channel (**c**).

Notably, the current structural studies were carried out on SUR2 subunits alone. However, our functional characterization of the $K_{ATP}$ channel in the presence of Kir6.2 (Fig. 6b) indicates the $K_{ATP}$ channel complex operates in the same manner as the SUR2 subunits. Moreover, we speculate that the dimerization of SUR2 activates the $K_{ATP}$ channel not only by extruding the inhibitory KNtp (Kir6 N-terminus) from its central vestibule but also by lowering the potency of the inhibitory ATP, similar to the activation of pancreatic SUR1-containing $K_{ATP}$ channel[33].

Taken together, our structural and functional analysis of SUR2A and SUR2B provides insights into their regulatory mechanisms and paves the way for further in-depth mechanistic studies.

# Methods

## Cell lines
FreeStyle 293F (Thermo Fisher Scientific # R79007) suspension cells were cultured in 293 Expression Medium (Gibco) supplemented with 1% FBS at 37 °C, with 6% $CO_2$ and 70% humidity. Sf9 insect cells (Thermo Fisher Scientific # 12659017) were cultured in Sf-900 III SFM medium (Thermo Fisher Scientific) at 27 °C. Cells obtained from vendors were not further authenticated.

## Expression constructs
cDNAs of SUR2A and SUR2B from *Rattus norvegicus* were cloned into C-terminal GFP-tagged BacMam expression vector which also contains two Strep tags. For functional studies, Kir6.2 was cloned into a modified C-terminal GFP-tagged BacMam expression vector and SUR2 mutants were cloned into a BacMam expression vector without tags as described previously[26].

## Electrophysiology
$K_{ATP}$ constructs were transfected into FreeStyle 293-F cells using polyethylenimine at a cell density of $1 \times 10^6$ cells/ml. Cells were cultured in FreeStyle 293 Expression Medium with 1% FBS for 24–36 h before recording. Macroscopic currents were recorded using inside-out mode at +60 mV in the pipette (membrane potential of −60 mV) through and an Axon-patch 200B amplifier (Axon Instruments, USA). Patch electrodes were pulled by a horizontal micro-electrode puller (P-1000, Sutter Instrument Co, USA) to tip resistance of 1.0–3.0 MΩ.

Pipette solution containing (mM): 140 KCl, 1.2 $MgCl_2$, 2.6 $CaCl_2$, 10 HEPES (pH 7.4, NaOH) and bath solution containing (mM): 140 KCl, 10 EGTA, 1 $MgCl_2$, 10 HEPES (pH 7.4, NaOH) were used for measuring inhibitory effect of Mg-ATP. For measuring the inhibitory effects of ATP without $Mg^{2+}$, bath solution containing (mM): 140 KCl, 10 EDTA, and 10 HEPES (pH 7.4, NaOH) was used. Signals were acquired at 5 kHz and low-pass filtered at 1 kHz. Data were further analyzed with pClampfit 10.0 software.

## Expression and purification of SUR2 subunits
SUR2 subunits were expressed using the BacMam system as described previously with minor modifications[23]. Briefly, cells were harvested 48 h post-infection and membrane pellets were purified as described previously[26]. For purification, membrane pellets were homogenized in TBS (20 mM Tris and 200 mM NaCl) and then solubilized in 1% GDN and 0.05% CHS for 30 min at 4 °C. Unsolubilized material was removed by centrifugation at $100,000 \times g$ for 30 min. The supernatant was supplemented with 1 mM ATP and 1 mM $MgCl_2$ and loaded onto Strep-tactin Beads 4FF (Smart Lifesciences). The beads were washed with buffer A (TBS with 50 μM GDN and 1 mM ATP) plus 10 mM $MgCl_2$ and protein was eluted with buffer A plus 10 mM desthiobiotin. GFP tags were removed by PreScission protease. To purify the SUR2A protein, proteins were concentrated by 100-kDa cut-off concentrator (Sartorius) and loaded onto Superose 6 increase (GE Healthcare) running in TBS with 50 μM GDN, 1 mM ATP. SUR2B is purified similarly to SUR2A.

## Cryo-EM sample preparation
Purified SUR2A proteins were further supplemented with 3 mM fluorinated Fos-Choline-8 (FFC), 3 mM $MgCl_2$, 2 mM ATP (Sigma) and 200 μM RPG (Abcam) for Mg-ATP + RPG samples; 3 mM $MgCl_2$, 2 mM ADP (Sigma) and 200 μM RPG for Mg-ATP/Mg-ADP samples. SUR2B samples are prepared in the same way as SUR2A. Cryo-EM sample was loaded on to glow-discharged Quantifoil 0.6/1 gold grids and frozen as described previously[23].

## Cryo-EM data acquisition
Cryo-grids were screened on Talos Arctica microscope (Thermo Fisher Scientific) operated at 200 kV and grids in good quality were transferred into Titan Krios microscope (Thermo Fisher Scientific) operated

at 300 kV for data acquisition. Images were collected using K3 camera (Gatan) mounted post a Quantum energy filter with 20 eV slit and operated under super-resolution mode with a pixel size of 0.834 Å at the object plane. Defocus values were set to range from −1.8 μm to −2.0 μm for data collection. Data were acquired by EPU-2.9.0.1519REL. The dose rate on the detector was 17.3 $e^- s^{-1} A^{-2}$. And the total exposure was 52 $e^- A^{-2}$. Each 3 s movie was dose-fractioned into 32 frames.

## Image processing
Collected movies were gain-corrected, motion-corrected, exposure-filtered, mag-distortion-corrected and binned with MotionCor2-1.3.2[45], producing dose-weighted, and summed micrographs with pixel size 0.834 Å. CTF models of dose-weighted micrographs were determined using GCTF-1.18[46]. Auto-picking was done by Gautomatch-0.56 (developed by Kai Zhang, MRC-LMB). Auto-picked particles were extracted from dose-weighted micrographs by a binning factor of 2. Datasets were subjected to 2D classification using RELION 3.0[47]. Particles yielding from 2D classification were subjected to 50 iterations $K = 1$ global search 3D classification with an angular sampling step of 7.5° to determine the initial alignment parameters using initial model generated by cryoSPARC-3.1.0[48] as reference. $K = 4$ multi-reference local angular search 3D classification was performed with an angular sampling step of 3.75° and a search range of 15°. The multi-references were generated using the initial model low-pass filtered to 8, 15, 25, and 35 Å, respectively. Particles from selected 3D classes were re-centered and re-extracted from summed micrographs to yield the pixel size of 0.834 Å. Particles were subjected to another round of initial model generation with $n = 3$ using cryoSPARC-3.1.0[49]. Particles in good classes were further refined against the initial model using non-uniform refinement and local refinement in cryoSPARC-3.1.0 to reach 3.0 Å (SUR2A$_{IF/MgATP/MgATP}$ state), 3.8 Å (SUR2A$_{IF/MgATP/MgADP}$ state), respectively. The SUR2A$_{IF/MgATP/MgATP}$ data was further subjected to seed-facilitated 3D classification[50] and non-uniform refinement, resulting in an increased particle number and a 2.84 Å final reconstruction. The data of SUR2B is processed similarly as SUR2A, except that particles were picked using Topaz-0.2.3[51]. After seed-facilitated 3D classification[50] and non-uniform refinement in cryoSPARC-3.1.0, particles selected from good classes were refined to 3.73 Å (SUR2B$_{IF/MgATP/MgATP}$ state), 3.57 Å (SUR2B$_{IF/MgATP/MgADP}$ state), 3.61 Å (SUR2B$_{PO/MgATP/MgADP}$ state), respectively.

## Model building
Maps were converted to MTZ files by PHENIX-1.18rc1-3777[52]. We used structures of the SUR1 subunit (6JB1) of our previous K$_{ATP}$ structure as the initial model. The homologous SUR2 structures were generated using SWISS-MODEL[53] and docked into the cryo-EM map with UCSF Chimera-1.14[54]. Models were manually rebuilt in Coot-0.9.2[55] and further refined by PHENIX-1.18rc1-3777[52]. The residues contained in the final models were indicated in Supplementary Table 1. Figures were prepared with Pymol-1.7.0.5 (Schrodinger, LLC.) and UCSF ChimeraX-0.91[55].

## Docking of the R helix
Coordinates of the R helix were isolated from the SUR2A$_{IF/MgATP/MgATP}$ structure and were docked onto the rest of the SUR2A$_{IF/MgATP/MgATP}$ structure or the SUR2A$_{IF/MgATP/MgADP}$ structure using the HDOCK server[37]. Docking results were ranked according to a knowledge-based iterative scoring function ITScorePP or ITScorePR and the top ten ranked structures were analyzed. Docked structures with the R helix-bound inside the central vestibule of SUR2A were rejected because in this pose, the R helix (part of the NBD1-TMD2 linker) could not connect to NBD1 and TMD2 anymore.

## Quantification and statistical analysis
Global resolution estimations of cryo-EM density maps are based on the 0.143 Fourier Shell Correlation criterion[56]. The local resolution map was calculated using cryoSPARC-3.1.0[49]. Electrophysiological data

reported were analyzed with pClampfit 10.0 software, calculated with Microsoft Excel and GraphPad Prism 5.0. The number of biological replicates (N) and the relevant statistical parameters for each experiment (such as mean or standard error) are described in figure legends. No statistical methods were used to pre-determine sample sizes.

## Reporting summary
Further information on research design is available in the Nature Portfolio Reporting Summary linked to this article.

## Data availability
The data that support this study are available from the corresponding authors upon request. Cryo-EM maps have been deposited in the Electron Microscopy Data Bank (EMDB) under accession codes EMD-33563 (SUR2A$_{IF/MgATP/MgATP}$), EMD-33564 (SUR2A$_{IF/MgATP/MgADP}$), EMD-33565 (SUR2B$_{IF/MgATP/MgATP}$), EMD-33566 (SUR2B$_{IF/MgATP/MgADP}$), and EMD-33567 (SUR2B$_{PO/MgATP/MgADP}$). Atomic coordinates have been deposited in the Protein Data Bank (PDB) under accession codes 7Y1J (SUR2A$_{IF/MgATP/MgATP}$), 7Y1K (SUR2A$_{IF/MgATP/MgADP}$), 7Y1L (SUR2B$_{IF/MgATP/MgATP}$), 7Y1M (SUR2B$_{IF/MgATP/MgADP}$), and 7Y1N (SUR2B$_{PO/MgATP/MgADP}$). 6JB1, 7VLU and 7VLS are available on PDB. Source data are provided with this paper.

## Materials availability
Reagents generated in this study will be made available on request, but we may require payment and/or a completed Materials Transfer Agreement if there is potential for commercial application.

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

## Acknowledgements

We thank Dr. Susumu Seino for sharing rnSUR2A and 2B cDNAs, J. Marc Simard for sharing mmKir6.2 cDNA. We thank Dr. Yunlu Kang for the advice on image processing. We thank the National Center for Protein Sciences at Peking University in Beijing, China for assistance with negative stain EM. Cryo-EM data collection was supported by Electron microscopy laboratory and Cryo-EM platform of Peking University with the assistance of Xuemei Li, Changdong Qin, Xia Pei, Xiaojuan Hui, Zhenxi Guo, and Guopeng Wang. Part of structural computation was also performed on the Computing Platform of the Center for Life Science (Life Science No. 1) and High-performance Computing Platform of Peking University (Wei-Ming No.1). The work is supported by grants from the Ministry of Science and Technology of China (National Key R&D Program of China, 2022YFA0806504 to L.C.) the National Natural Science Foundation of China (91957201, 32225027 and 31821091 to L.C.), and Center For Life Sciences (CLS).

## Author contributions

L.C. initiated the project. D.D. and T.H. purified protein, prepared the cryo-EM sample, and performed electrophysiology experiments. D.D., T.H., M.W., and J.-X.W. collected the data. D.D. and T.H. processed the data, built and refined the model. All authors contributed to the manuscript preparation.

## Competing interests

The authors declare no competing interests.
