## [Peer Review File · Nature Communications]

The inhibition mechanism of the SUR2A-containing KATP channel by a regulatory helixReviewers' Comments:

Reviewer #1:

Remarks to the Author:

KATP channel senses the intracellular ATP/ADP ratio via regulatory ABC proteins such as SUR1 and SUR2. This manuscript described multiple cryo-EM structures of rat SUR2 with an allosteric inhibitor RPG under different conditions of ATP/ADP. The authors discovered a previously-overlooked regulatory helix (R-helix) residing in between the NBDs. This R-helix functions as an inhibitory regulator as demonstrated by structural, mutagenesis, as well as electrophysiological experiments. If the result is true, it will surely advance our understanding about functional regulation of KATP channels. However, the reviewer's enthusiasm dampens by the overinterpretation of cryo-EM data, as detailed later. Thus, a major revision seems necessary though the finding of the manuscript is very interesting.

1. Fig 1: there's no indication regarding the position of RPG in the EM map. In other words, how are panels a and e connected? It is also better to change the color of RPG to be different from R helix.
2. Line 112: the description of the model is not clear. It should be indicated in the text (and probably in the method section too) that what residues are missing in the model. Since the R-helix is in the middle of a flexible loop and most part of the loop is invisible, this information will be critical for the readers to judge the quality of the structure.
3. Line 119: "R930 interacts electrostatically with D789" is a bit ambiguous. From Fig 2B, it looks like the side chains of D789 and R930 are opposing instead of attracting to each other.
4. Line 133: The authors claim 2 structures reconstructed from SUR2A+Mg-ATP/Mg-ADP+RPG dataset, and the difference is that NBD2 binds either ADP or ATP. Personally, I couldn't see such difference from the cryo-EM densities provided in extended Fig 2 and 3. The quality of the densities in extended Fig 3 (g, h) is so poor and the authors shouldn't be able to distinguish ATP/ADP. In addition, extended Fig 3c shows clearly that the local resolution of NBDs is lower than 5.5Å, which is way too low to confirm any small molecule densities.
5. Fig 3: the comparison of the 2 structures should be quantified by parameters such as RMSD. By simply looking at it, I would say they are almost the same and the comparison is probably meaningless.
6. Extend Fig 4, Line 144: The authors claim that NBD2 is highly flexible because one 3D class does not have the domain. This is over interpretation of cryo-EM data, as we know that many particles are simply damaged during the freezing and imaging process. The missing NBD is likely just averages of damaged particles. In addition, the best reconstruction is ~3.7Å from this dataset while the first dataset reaches ~3Å, suggesting that the quality of this dataset is simply not that great.
7. Line 160: SUR2BOD/MgATP/MgADP is the one appeared in Line 83, but with different names.
8. Line 164: The authors claim that "The binding of RPG precludes the fully closure of TMD". Where does RPG binds in the TMD is not clearly stated. Does RPG bind to the similar site as other KCO molecules used in previous studies? To make such a claim, it is important to give such information.
9. Line 220: About C42, where the corresponding density is in SUR2 has never been clearly marked, though the model is built. Thus, it is hard for the readers to judge any statement about this domain.

Reviewer #2:

Remarks to the Author:

Information regarding the structure, function and regulation of the ATP-sensitive potassium channel family is fraught with difficulties. The channels are ubiquitous, but their roles are often poorly understood, the pharmacology is incomplete, and the field is plagued with some rather poor

information. It is, therefore, welcome that in recent years there have been some strides made to understand the regulation of this channel at the molecular level. This is an example of one such manuscript. This is one of the first publications to use the SUR2 accessory subunit, where much work has been focussed on the SUR1 subunit to date. The cryo-EM structures used in this study add to the growing body of work that is starting to uncover some of the mysteries surrounding this widely-studied, but poorly understood channel complex. It is now widely accepted that the motility of the subunits within the KATP complex is vast, perhaps unsurprising given the size of the accessory subunits. The motility of each subunit has hindered some of the structural analyses from various groups, along with the vast size of the channel complex within the membrane. The authors here have approached this in a somewhat different way using purified SUR2 subunits rather than using complete channel complexes.

There are some interesting outcomes from this study. Firstly, the ability to resolve the R helix that is not been seen in other SUR structures to date. It would be interesting to see whether this helix is still present within the full KATP structure, or if this is a unique feature of a misfolding of the expressed protein. This helix is present in other ABC proteins, as the authors state, and so this is an interesting finding. Furthermore, the binding of ADP causing this R helix to shift gives a strong indication of a structural rearrangement on binding of ADP that could lead to a conformational change to activate the open pore configuration of the channel complex. Furthermore, the slightly different things occurring in the SUR2A vs 2B also give further weight to their findings as SUR2A and 2B respond differently to ATP, ADP and pharmacological modulators despite there only being a small difference in the C-terminal length of 42 residues. This evidence suggests that there is a structural difference between the two subunits despite this small extra c-terminal segment being the only difference between the two splice variants.

1) It would be helpful for the authors to discuss how the structural motifs and arrangements identified in their cryo-EM data correlate with the SUR1 propellor and quatrefoil arrangements outlined in other recent studies for SUR1. Are there structural similarities between the arrangements of SUR1 and SUR2 in terms of their gross structure? Given the marked differences in properties, I think that this comparison between SUR1 and 2 would be helpful in this study.

2) I would have liked to have read more about the R-helix, in terms of its conservation across the ABC family. You've discussed CFTR, but also stated that the R-helix was not observed in SUR2B-Kir6.1, perhaps due to poor resolution at that region. Is this why it has not been observed in SUR1? Has SUR1 been measured in the same manner that you have done or has it solely been in complex with Kir6.2? Is the presence of the Kir6.2 constraining other parts of the molecule that makes this R-helix region more motile and so more difficult to resolve?

3) I am confused by extended figure 9 – are the currents shown in this the same data as in figure 6 (I think they are)? If so this needs clarification. Could the authors also please show where the zero level of activity is with a dotted line on each trace so that the non-expert can appreciate that the block with 1 mM ATP is usually complete for the Kir6.2/SUR2A complex. Your currents were recorded at +60 mV; has this been accounted for in the figure traces? Recording at +60 mV pipette potential is the equivalent of -60 mV in the patch, therefore the currents recorded would be in the inward, not outward direction (for symmetrical potassium solutions as indicated in the methods section)? If this is true then the traces need inverting to represent the conventional display of outward and inward currents. Recording at -60 mV would be more appropriate given the weak rectifier properties of Kir6.2.

4) Some of the introduction discussing the expression of the subunits is a little out of date. SUR1, 2A, and 2B have been shown to be expressed in different regions of the heart alone.

5) Some of the text needs polishing; e.g. Line 42 "splicing isoforms", perhaps "splice variants", Line 45 "skeleton muscle", perhaps "skeletal".

Reviewer #3:

Remarks to the Author:

The relatively high impact molecular mechanistic result of the R-helix docking between NBDs as a regulatory switch in KATP channels is reported here with significant implications for human physiology.

The best model/map showing this helix is marginally convincing and could be more so with some additional evidence, as discussed below.

The expression, purification and cryoEM approach and data appear sound, the manuscript is well organized and concise and there is some functional data that appears consistent with the structural findings, however, the main question is whether the key observation of the R-helix is biologically relevant/real or to what degree it is. Notably, since the structure is solved without Kir channel and the functional measurements are made with the full KATP channel, the connection between the two needs to be strengthened. In particular, the disordered TMD0 is the conduit for SUR regulation of the Kir pore, so there is every possibility that these structures may not represent the cellular context. To this end, further work should be done:

1. The mutational analysis is not sufficiently convincing. since inside-out patch clamping was done, the r-helix as an isolated peptide can be applied at increasing doses to inhibit currents in situ, as would be predicted by the model. This would help convince that the R-helix model is real
2. SUR2B should be co-expressed with Kir6.1 in the electrophysiology measurements. Kir6.2/SUR2B is not a common complex and may be more unstable than the other Kir/SUR combinations. This could be an alternative explanation for the SUR2B phenotypes and structures. Accordingly, functional measurements should include Kir6.1/SURB to draw mechanistic conclusions.
3. the R-helix should be either docked by computational molecular docking (conformational search and energy calculation) either as a rigid body with flexible side chains or as a fully flexible peptide to establish its biophysical compatibility with its docking site. Other data demonstrating that it does not clash/fits well via van der Waals and electrostatics could also suffice.

Methodologically there are some potential improvements

1. SWISS-MODEL is not state of the art for homology modeling, potentially leading to errors in the model. More sophisticated homology models, e.g. AlphaFold2, should be docked into the maps to generate the models.
2. Some geometric validation data would be helpful such as Procheck or ERRAT especially for the C42 segment

We are grateful for the constructive suggestions from editors and reviewers. Please find the point-to-point responses below.

Reviewer #1 (Remarks to the Author):

KATP channel senses the intracellular ATP/ADP ratio via regulatory ABC proteins such as SUR1 and SUR2. This manuscript described multiple cryo-EM structures of rat SUR2 with an allosteric inhibitor RPG under different conditions of ATP/ADP. The authors discovered a previously-overlooked regulatory helix (R-helix) residing in between the NBDs. This R-helix functions as an inhibitory regulator as demonstrated by structural, mutagenesis, as well as electrophysiological experiments. If the result is true, it will surely advance our understanding about functional regulation of KATP channels. However, the reviewer's enthusiasm dampens by the overinterpretation of cryo-EM data, as detailed later. Thus, a major revision seems necessary though the finding of the manuscript is very interesting.

1. Fig 1: there's no indication regarding the position of RPG in the EM map. In other words, how are panels a and e connected? It is also better to change the color of RPG to be different from R helix.

Response: We have revised Fig. 1, Supplementary Fig. 3i, Supplementary Fig. 5c, Supplementary Fig. 7e, and Supplementary Fig. 8e for a better illustration of the position of the RPG.

2. Line 112: the description of the model is not clear. It should be indicated in the text (and probably in the method section too) that what residues are missing in the model. Since the R-helix is in the middle of a flexible loop and most part of the loop is invisible, this information will be critical for the readers to judge the quality of the structure.

Response: We have included detailed information in the revised Supplementary Table 1. We have also provided revised Fig. 1e for better illustration of unmodeled residues.

3. Line 119: "R930 interacts electrostatically with D789" is a bit ambiguous. From Fig 2B, it looks like the side chains of D789 and R930 are opposing instead of attracting to each other.

Response: In the structure, we found that D789 interacts with both R815 and R930. We have provided a revised Fig. 2b for better visualization of the interaction between D789 and R930.

4. Line 133: The authors claim 2 structures reconstructed from SUR2A+Mg-ATP/Mg-ADP+RPG dataset, and the difference is that NBD2 binds either ADP or ATP. Personally, I couldn't see such difference from the cryo-EM densities provided in extended Fig 2 and 3. The quality of the densities in extended Fig 3 (g, h) is so poor and the authors shouldn't be able to distinguish ATP/ADP. In addition, extended Fig 3c shows clearly that the local resolution of NBDs is lower than 5.5Å, which is way too low to confirm any small molecule densities.

Response: We agree with this reviewer that the identity of the nucleotides could not be assigned purely based on the electron density map. Based on our previous studies (Ding et al., Nat Commun, 2022, PMID: 35562524), which showed that NBD1 was always bound with Mg-ATP even in the presence of a high concentration of Mg-ADP, we assign the nucleotide density at NBD1 as Mg-ATP. Moreover, because of the different conformation of the second 3D class compared with the first 3D class (SUR2A_{IF/MgATP/MgATP}), we reasoned that Mg-ADP rather than Mg-ATP should bind at NBD2, otherwise the second 3D class would have the same structure as SUR2A_{IF/MgATP/MgATP}. Therefore, we assign the second 3D class as SUR2A_{IF/MgATP/MgADP}. We have included these discussions from line 144 to line 151.

5. Fig 3: the comparison of the 2 structures should be quantified by parameters such as RMSD. By simply looking at it, I would say they are almost the same and the comparison is probably meaningless.

Response: We have included the RMSD value in Fig. 3. The major difference between these two structures is we could observe R helix density in SUR2A_{IF/MgATP/MgATP}, but not in SUR2A_{IF/MgATP/MgADP}, suggesting the flexibility of R helix in this structure.

6. Extend Fig 4, Line 144: The authors claim that NBD2 is highly flexible because one 3D class does not have the domain. This is over interpretation of cryo-EM data, as we know that many particles are simply damaged during the freezing and imaging process. The missing NBD is likely just averages of damaged particles. In addition, the best reconstruction is $\sim 3.7\text{\AA}$ from this dataset while the first dataset reaches $\sim 3\text{\AA}$, suggesting that the quality of this dataset is simply not that great.

Response: We agree with the reviewer that it is possible that this class represents damaged particles as indicated in the revised manuscript at line 157-162. But we also want to emphasize that under the same cryo-EM sample preparation condition, we did not observe such 3D classes in the sample of SUR2A, suggesting the flexibility of NBD2 might be an endogenous property of SUR2B. Similar phenomenon has also been observed recently in the structures of $\Delta 508$ CFTR (Fiedorczuk and Chen, Science, 2022, PMID: 36264792): when correctors were absent, one NBD of CFTR is disordered.

7. Line 160: SUR2B_{OD/MgATP/MgADP} is the one appeared in Line 83, but with different names.

Response: We have changed the expression in line 83 into “SUR2A_{OD/MgATP/MgADP} and SUR2B_{OD/MgATP/MgADP}”.

8. Line 164: The authors claim that “The binding of RPG precludes the fully closure of TMD”.

Where does RPG binds in the TMD is not clearly stated. Does RPG bind to the similar site as other KCO molecules used in previous studies? To make such a claim, it is important to give such information.

Response: We have provided the position of RPG binding sites in the revised Fig. 1a, d. Both KCO and RPG bind to the TMD of SUR but at different locations with 20 Å away from each other, as detailed in our previous studies (Ding et al., Cell Rep, 2019, PMID: 31067468; Ding et al., Nat Commun, 2022, PMID: 35562524; Wu et al., Mol. Pharmacol., 2022, PMID: 36253099).

9. Line 220: About C42, where the corresponding density is in SUR2 has never been clearly marked, though the model is built. Thus, it is hard for the readers to judge any statement about this domain.

Response: We have provided the location of C42 in Supplementary Fig. 1k, l.

Reviewer #2 (Remarks to the Author):

Information regarding the structure, function and regulation of the ATP-sensitive potassium channel family is fraught with difficulties. The channels are ubiquitous, but their roles are often poorly understood, the pharmacology is incomplete, and the field is plagued with some rather poor information. It is, therefore, welcome that in recent years there have been some strives made to understand the regulation of this channel at the molecular level. This is an example of one such manuscript. This is one of the first publications to use the SUR2 accessory subunit, where much work has been focussed on the SUR1 subunit to date. The cryo-EM structures used in this study add to the growing body of work that is starting to uncover some of the mysteries surrounding this widely-studied, but poorly understood channel complex. It is now widely accepted that the motility of the subunits within the KATP complex is vast, perhaps unsurprising given the size of the accessory subunits. The motility of each subunit has hindered some of the structural analyses from various groups, along with the vast size of the channel complex within the membrane. The authors here have approached this in a somewhat different way using purified SUR2 subunits rather than using complete channel complexes.

There are some interesting outcomes from this study. Firstly, the ability to resolve the R helix that is not been seen in other SUR structures to date. It would be interesting to see whether this helix is still present within the full KATP structure, or if this is a unique feature of a misfolding of the expressed protein. This helix is present in other ABC proteins, as the authors state, and so this is an interesting finding. Furthermore, the binding of ADP causing this R helix to shift gives a strong indication of a structural rearrangement on binding of ADP that could lead to a conformational change to activate the open pore configuration of the channel complex.

Furthermore, the slightly different things occurring in the SUR2A vs 2B also give further weight to their findings as SUR2A and 2B respond differently to ATP, ADP and pharmacological modulators despite there only being a small difference in the C-terminal length of 42 residues. This evidence suggests that there is a structural difference between the two subunits despite this

small extra c-terminal segment being the only difference between the two splice variants.

1) It would be helpful for the authors to discuss how the structural motifs and arrangements identified in their cryo-EM data correlate with the SUR1 propeller and quatrefoil arrangements outlined in other recent studies for SUR1. Are there structural similarities between the arrangements of SUR1 and SUR2 in terms of their gross structure? Given the marked differences in properties, I think that this comparison between SUR1 and 2 would be helpful in this study.

Response: Our current work has been done using isolated SUR2A or SUR2B subunits and we do not have evidence to support the previously observed propeller or quatrefoil structures. We have included the structural comparison between SUR1 and SUR2 in Supplementary Fig. 1m.

2) I would have liked to have read more about the R-helix, in terms of its conservation across the ABC family. You've discussed CFTR, but also stated that the R-helix was not observed in SUR2B-Kir6.1, perhaps due to poor resolution at that region. Is this why it has not been observed in SUR1? Has SUR1 been measured in the same manner that you have done or has it solely been in complex with Kir6.2? Is the presence of the Kir6.2 constraining other parts of the molecule that makes this R-helix region more motile and so more difficult to resolve?

Response: We have determined the structure of the SUR1 subunit in complex with mitiglinide (Wang et al., *Front Pharmacol*, 2022, PMID: 35847046) but did not observe the R helix. We think this is because the R helix is mutated in SUR1: there is a six-residue insertion at the corresponding region of SUR1 as shown in Fig. 2d. Such insertion would greatly decrease the inhibition of the R helix on SUR2A, as shown in Fig. 6a.

3) I am confused by extended figure 9 – are the currents shown in this the same data as in figure 6 (I think they are)? If so this needs clarification. Could the authors also please show where the zero level of activity is with a dotted line on each trace so that the non-expert can appreciate that the block with 1 mM ATP is usually complete for the Kir6.2/SUR2A complex. Your currents were recorded at +60 mV; has this been accounted for in the figure traces? Recording at +60 mV pipette potential is the equivalent of -60 mV in the patch, therefore the currents recorded would be in the inward, not outward direction (for symmetrical potassium solutions as indicated in the methods section)? If this is true then the traces need inverting to represent the conventional display of outward and inward currents. Recording at -60 mV would be more appropriate given the weak rectifier properties of Kir6.2.

Response: We have added the zero level of currents to Supplementary Fig. 9. We used +60 mV in the pipette to mimic the negative potential inside the cell as indicated in line 302 in the method section. The same protocol has been used in other study (Sung et al., *J. Mol. Biol.*, 2022, PMID: 35964676).

4) Some of the introduction discussing the expression of the subunits is a little out of date. SUR1, 2A, and 2B have been shown to be expressed in different regions of the heart alone.

Response: We have revised the expression from line 43-46 and cited a more recent review paper by (Nichols, Card Electrophysiol Clin, 2016, PMID: 27261824).

5) Some of the text needs polishing; e.g. Line 42 “splicing isoforms”, perhaps “splice variants”, Line 45 “skeleton muscle”, perhaps “skeletal”.

Response: We have made these modifications in the revised manuscript.

Reviewer #3 (Remarks to the Author):

The relatively high impact molecular mechanistic result of the R-helix docking between NBDs as a regulatory switch in KATP channels is reported here with significant implications for human physiology. The best model/map showing this helix is marginally convincing and could be more so with some additional evidence, as discussed below.

The expression, purification and cryoEM approach and data appear sound, the manuscript is well organized and concise and there is some functional data that appears consistent with the structural findings, however, the main question is whether the key observation of the R-helix is biologically relevant/real or to what degree it is. Notably, since the structure is solved without Kir channel and the functional measurements are made with the full KATP channel, the connection between the two needs to be strengthened. In particular, the disordered TMD0 is the conduit for SUR regulation of the Kir pore, so there is every possibility that these structures may not represent the cellular context. To this end, further work should be done:

1. The mutational analysis is not sufficiently convincing. since inside-out patch clamping was done, the r-helix as an isolated peptide can be applied at increasing doses to inhibit currents in situ, as would be predicted by the model. This would help convince that the R-helix model is real.

Response: As suggested by the reviewer, we have synthesized the R-helix peptide “DQTTLERKTLRRAMYSR” and applied 100 μ M of this peptide to the inside-out patch containing SUR2A (R930A+Y838A)/Kir6.2 channel for the trans-inhibition assay. But unfortunately, we did not find any inhibition as shown below. We reasoned this is because when the R helix is tethered to SUR2A, the R helix is restricted inside a local sphere with a diameter of 40 Å. The local concentration of the R helix is about 6.2 M, which is far more than the concentration we could apply to the patch in trans.

2. SUR2B should be co-expressed with Kir6.1 in the electrophysiology measurements. Kir6.2/SUR2B is not a common complex and may be more unstable than the other Kir/SUR combinations. This could be an alternative explanation for the SUR2B phenotypes and structures. Accordingly, functional measurements should include Kir6.1/SUR2B to draw mechanistic conclusions.

Response: The expression of the Kir6.2-SUR2B combination has been implicated in coronary artery (Yoshida et al., J. Mol. Cell. Cardiol., 2004, PMID: 15380676), demonstrating the physiological relevance of this combination. Moreover, we observed robust K_{ATP} current using this combination (Supplementary Fig. 9), suggesting the complex is stable on the membrane. Furthermore, we have also tried to measure the inside-out currents of Kir6.1-SUR2B and found the currents were much smaller than Kir6.2-SUR2B. Therefore, to get a high signal-to-noise ratio, we used Kir6.2-SUR2B for electrophysiology measurement. In addition, to be a fair comparison with the Kir6.2-SUR2A channel, we used the Kir6.2-SUR2B, in case the different properties between Kir6.1 and Kir6.2 would lead to potential bias.

3. the R-helix should be either docked by computational molecular docking (conformational search and energy calculation) either as a rigid body with flexible side chains or as a fully flexible peptide to establish its biophysical compatibility with its docking site. Other data demonstrating that it does not clash/fits well via van der Waals and electrostatics could also suffice.

Response: We have provided revised Fig. 3d to show the sterical clashes between R helix and SUR2A_{IF}/MgATP/MgADP.

Methodologically there are some potential improvements

1. SWISS-MODEL is not state of the art for homology modeling, potentially leading to errors in the model. More sophisticated homology models, e.g. AlphaFold2, should be docked into the maps to generate the models.

Response: When we built the models, the AlphaFold 2 was not available yet. Retrospectively, we compared our model with the model predicted by AlphaFold 2 and found their registration was consistent.

2. Some geometric validation data would be helpful such as Procheck or ERRAT especially for the C42 segment.

Response: We have checked the geometric quality of C42 using MolProbity integrated in Phenix suite, and the models were of reasonable quality.

References:

- Ding, D., Wang, M., Wu, J.X., Kang, Y., and Chen, L. (2019). The Structural Basis for the Binding of Repaglinide to the Pancreatic KATP Channel. *Cell Rep* 27, 1848-1857 e1844.
- Ding, D., Wu, J.X., Duan, X., Ma, S., Lai, L., and Chen, L. (2022). Structural identification of vasodilator binding sites on the SUR2 subunit. *Nat Commun* 13, 2675.
- Fiedorczuk, K., and Chen, J. (2022). Molecular structures reveal synergistic rescue of Delta508 CFTR by Trikafta modulators. *Science* 378, 284-290.
- Nichols, C.G. (2016). Adenosine Triphosphate-Sensitive Potassium Currents in Heart Disease and Cardioprotection. *Card Electrophysiol Clin* 8, 323-335.
- Sung, M.W., Driggers, C.M., Mostofian, B., Russo, J.D., Patton, B.L., Zuckerman, D.M., and Shyng, S.L. (2022). Ligand-mediated Structural Dynamics of a Mammalian Pancreatic KATP Channel. *J. Mol. Biol.* 434, 167789.
- Wang, M., Wu, J.X., and Chen, L. (2022). Structural Insights Into the High Selectivity of the Anti-Diabetic Drug Mitiglinide. *Front Pharmacol* 13, 929684.
- Wu, J.X., Ding, D., and Chen, L. (2022). The Emerging Structural Pharmacology of ATP-Sensitive Potassium Channels. *Mol. Pharmacol.* 102, 234-239.
- Yoshida, H., Feig, J.E., Morrissey, A., Ghiu, I.A., Artman, M., and Coetzee, W.A. (2004). K ATP channels of primary human coronary artery endothelial cells consist of a heteromultimeric complex of Kir6.1, Kir6.2, and SUR2B subunits. *J. Mol. Cell. Cardiol.* 37, 857-869.

Reviewers' Comments:

Reviewer #1:

Remarks to the Author:

The authors partially answered my questions. However, my concern about the "over-interpretation" is not alleviated. Several serious questions remain:

1. What is the quality of the experimental density of R-helix? How do the authors make sure the density observed is indeed the R-helix?
2. Reviewer #3 suggested an excellent experiment using R-helix peptides. With the available R-helix peptide, did the authors at least test the binding with the protein? Or further, structural determination of the complex of peptide/protein (without R-helix) would be an excellent way to answer my first question.
3. I suggested structural comparison using parameters such as RMSD. The authors did include one word of RMSD in line 259. In Figure 3, RMSD=1.177 is shown. Such a small number suggests that the two structures are primarily identical, reinforcing the idea of over-interpretation.

Reviewer #2:

Remarks to the Author:

The authors have addressed all of my previous comments.

Reviewer #3:

Remarks to the Author:

Regarding this revision, the responses improve the manuscript but still leave some doubt as to the confidence of the R-helix conformation. The authors explanation of local concentration for the R-helix obviating in vitro confirmation of its effect is acceptable as is their response to evaluating the Kir/SUR complex. Responses to the technical suggestions of AlphaFold and quality metrics are responsive as well.

However, in the absence of further in vitro data establishing the bioactivity of the R-helix the energetic evaluation of the R-helix docked in the proposed site becomes more important. Visual steric clashes cited in the response are not sufficient. Van der Waals, Coulomb electrostatics and solvation energy terms at a minimum for the published conformation should be provided and there is a problem if the overall energy is not significantly negative. Possibly, it should be shown that the R-helix self-docks (remove it from the model, dock it back) to this site with low energy (data on conformations searched and energy calculation should be provided). This will suggest that the R-helix prefers that groove over any other non-obvious site on the structure or indeed prefers that site to full solvation.

Ultimately, SOME further data is required for this to be convincing.

We are grateful for the constructive suggestions from editors and reviewers. During this revision, we have improved the map resolution of SUR2A_{IF/MgATP/MgATP} to 2.8Å resolution as shown in Supplementary Fig. 1. To make reviewers access the electron density and model easily, we have provided the download link for SUR2A_{IF/MgATP/MgATP} via dropbox (**please ask editor for map/model files**)

Because the reviewers concentrate their doubts on the confidence of the atomic model of the R helix, we would like to emphasize the experimental evidence that supports our atomic model: **First**, the connectivity of the electron density map (Supplementary Fig. 1m) and local map quality (Fig. 2a and Supplementary Fig. 1j) are sufficient to model the R helix, as shown in the electron density map provided in the revised manuscript. **Second**, the structure model of the R helix is thoroughly validated via mutagenesis experiments shown in Fig. 6. We think these data are sufficient to confirm the model and we could not find any other alternative explanation for both the R helix density and the mutagenesis data.

Please find the point-to-point responses in dark blue below.

Reviewer #1 (Remarks to the Author):

The authors partially answered my questions. However, my concern about the over-interpretation" is not alleviated. Several serious questions remain:

1. What is the quality of the experimental density of R-helix? How do the authors make sure the density observed is indeed the R-helix?

Response: The electron density map at the low contour level of SUR2A_{IF/MgATP/MgATP} clearly shows the continuous density between NBD1 and R helix, as shown in the revised Supplementary Fig. 1m, suggesting R helix is part of NBD1-TMD2 linker (913-985). The amino acids on this linker are QDQELEKDM EADQTTLERKTLRRAMYSREAKAQMEDEDEEEEEDEDDNMSTV MRLRTKMPWKTCWWYLTSG. Moreover, the map of the R helix is good enough for us to identify its identity as 924-940 (DQTTLERKTLRRAMYSR). We have provided the local density map of the R helix in Fig. 2a and Supplementary Fig. 1j and local resolution estimation in Supplementary Fig. 1f-h. In addition, we have also provided the final map and PDB of SUR2A_{IF/MgATP/MgATP} for reviewers' inspection. If the reviewer prefers other metrics to measure the quality of the experimental density of the R helix, we could provide such information. We are also happy to discuss if the reviewers think there is an alternative way to build the model for the density of the R helix.

2. Reviewer #3 suggested an excellent experiment using R-helix peptides. With the available R-helix peptide, did the authors at least test the binding with the protein? Or further, structural determination of the complex of peptide/protein (without R-helix) would be an excellent way to answer my first question.

Response: As shown in our last response, the isolated R helix at 100 μ M concentration could not inhibit the currents of the SUR2A-R930A-Y938A mutant probably due to the low affinity of the isolated R helix. This result also suggests that the Kd of the R helix to the SUR2A-R930A-Y938A mutant might be much higher than 100 μ M. Unfortunately, there is no reliable method to detect such low-affinity protein-protein interaction. We agree with the reviewer that the structural determination of the R helix deletion mutant would provide additional validation for the model of the R helix. However, as indicated in our response to question #1, the current map quality is already sufficient to support the model building of the R helix. Moreover, considering a large amount of time and effort to determine the structure of the artificial R helix-deletion mutant without high physiological significance, and the limited funding and resource, we have not carried out this experiment yet.

3. I suggested structural comparison using parameters such as RMSD. The authors did include one word of RMSD in line 259. In Figure 3, RMSD=1.177 is shown. Such a small number suggests that the two structures are primarily identical, reinforcing the idea of over-interpretation.

Response: In our last revision, we calculated the RMSD number in Fig. 3 using pymol with default parameters. It turns out that pymol would refine the aligned residues and exclude outlier residues to minimize the RMSD value. So we re-calculated the numbers without refinement and provided the number in the revised Fig. 3. The overall RMSD is 1.698 Å (8402 to 8402 atoms). This is consistent with the fact that both SUR2A_{IF/MgATP/MgADP} and SUR2A_{IF/MgATP/MgATP} are all in similar inward-facing conformation and their TMD could be aligned to some extent.

Next, we focused on the NBD layer where the R helix binds, and the RMSD of the NBD layer (668-913 of NBD1 and 1296-1542 of NBD2) is 2.204 Å (3188 to 3188 atoms), an indication for the structural difference. In addition, we have also provided distance measurements of marker atoms in the revised Fig. 3d-f, which clearly show the closure of NBDs in the SUR2A_{IF/MgATP/MgADP} state compared to that in the SUR2A_{IF/MgATP/MgATP} state.

Reviewer #2 (Remarks to the Author):

The authors have addressed all of my previous comments.

Reviewer #3 (Remarks to the Author):

Regarding this revision, the responses improve the manuscript but still leave some doubt as to the confidence of the R-helix conformation. The authors explanation of local concentration for the R-helix obviating in vitro confirmation of its effect is acceptable as is their response to evaluating the Kir/SUR complex. Responses to the technical suggestions of AlphaFold and quality metrics are responsive as well.

However, in the absence of further in vitro data establishing the bioactivity of the R-helix the energetic evaluation of the R-helix docked in the proposed site becomes more important. Visual steric clashes cited in the response are not sufficient. Van der Waals, Coulomb electrostatics and solvation energy terms at a minimum for the published conformation should be provided and there is a problem if the overall energy is not significantly negative. Possibly, it should be shown that the R-helix self-docks (remove it from the model, dock it back) to this site with low energy (data on conformations searched and energy calculation should be provided). This will suggest that the R-helix prefers that groove over any other non-obvious site on the structure or indeed prefers that site to full solvation.

Ultimately, SOME further data is required for this to be convincing.

Response: To validate the binding of the R helix, we have performed the docking calculation as suggested by the reviewer using the HDOCK server. We isolated the structure of the R helix (924-940) and docked it onto the remaining model of SUR2A_{IF/MgATP/MgATP} (see R-Fig. 1 below). Among the top 10 structures, one is the same as our model, with a docking score of -240.84 and the confidence score of 0.8073 (pose A). The docking score is calculated by a knowledge-based iterative scoring function ITScorePP or ITScorePR, which is related to the binding energy including Van der Waals, Coulomb electrostatics, and solvation energy terms. A more negative docking score means a more possible binding model. This suggests the binding of the R helix to the SUR2A_{IF/MgATP/MgATP} is energetically favorable. The confidence score is an empirically defined docking score-dependent score to indicate the binding likeliness of two molecules. When the confidence score is above 0.7, the two molecules would be very likely to bind. Among the top 10 structures, the other docking models placed the R helix inside the central vestibule of SUR2A (pose B). Pose B is physically impossible since the R helix is on the NBD1-TMD2 linker and its N and C termini should be exposed to solvent to connect with other parts of the protein. This computational docking result is consistent with our assignment of the R helix as 924-940 of SUR2A. The docking result is included in the revised manuscript line 116-121 and the procedures were provided in the method section.

As to Fig.3, we experimentally observed that the R helix is absent in the SUR2A_{IF/MgATP/MgADP} structure and we reasoned that this is because the conformational changes of the NBD layer lead to the sterical clashes between the R helix and NBDs (as shown in the revised Fig. 3g-h), therefore, the R helix could not bind to SUR2A_{IF/MgATP/MgADP}. In addition, we docked the R helix onto the structure of SUR2A_{IF/MgATP/MgADP}, we found all of the top 10 structures have the R helix in pose B, suggesting that the R helix could not bind in pose A anymore. This is consistent with our experimental observation showing that the R helix is not observed in the SUR2A_{IF/MgATP/MgADP} (revised Fig. 3a-b). This result is included in the revised manuscript line 159-162. We have also revised Fig. 3g-h to better illustrate several sterical clashes between the R

helix and SUR2A_{IF/MgATP/MgADP}. As aforementioned, these sterical clashes provide a strong explanation for the reason why the R helix could not bind to SUR2A_{IF/MgATP/MgADP}.

R-Fig. 1: Two docking poses of R helix (yellow or purple) on SUR2A_{IF/MgATP/MgATP} (green). Pose A is the same as our experimental structure. Pose B is not possible because of the physical connection of the R helix with NBD1 and TMD2.

Reviewers' Comments:

Reviewer #1:

Remarks to the Author:

I would like to thank the authors for providing the map of 2.8Å and the corresponding PDB file.

1. After careful examination, I am thrilled to see the nice fitting of the inhibitor RPG and nucleotides. However, I do not see the claimed "connection" between R-helix and NBD1. Based on my limited experience, I would be cautious about claiming that there's a "continuous density". Because if there is, I would build at least the backbone of the loop and show it to the readers. Additionally, in this revised manuscript, no direct binding experiment exists (wet lab) to show the interactions between the R-helix (peptide) and the transporter. Altogether, I am still not convinced that the density within NBDs is R-helix. It could just be impurities from protein purification/reagent.

2. For the RMSD values, the authors seem to be confused a bit. I was talking about the RMSD between SUR2AIF/MgATP/MgATP and SUR2AIF/MgATP/MgADP, not the difference between NBD1 and NBD2. In addition, although the RMSD values are shown in the figure, there is no description in the text to explain it, which is odd.

3. I am not convinced about the structure of SUR2AIF/MgATP/MgADP, especially the nucleotide-binding status. The authors "speculated" that the NBD2 is bound with MgADP because NBDs are not exactly the same as in SUR2AIF/MgATP/MgATP (and the first class of SUR2AIF/MgATP/MgADP). Such structural differences could probably be simply because of the resolution difference between the classes. Without the figure shown (supp fig3g,h), it is safe to say that the density is too bad to fit anything (or you can fit any nucleotides you like).

Overall, the rebuttal from the authors does not alleviate my concern about the "over-interpretation" of the data.

We are grateful for the constructive suggestions from editors and reviewers.

Please find the point-to-point responses in dark blue below.

REVIEWER COMMENTS

Reviewer #1 (Remarks to the Author):

I would like to thank the authors for providing the map of 2.8Å and the corresponding PDB file.

1. After careful examination, I am thrilled to see the nice fitting of the inhibitor RPG and nucleotides. However, I do not see the claimed “connection” between R-helix and NBD1. Based on my limited experience, I would be cautious about claiming that there’s a “continuous density”. Because if there is, I would build at least the backbone of the loop and show it to the readers. Additionally, in this revised manuscript, no direct binding experiment exists (wet lab) to show the interactions between the R-helix (peptide) and the transporter. Altogether, I am still not convinced that the density within NBDs is R-helix. It could just be impurities from protein purification/reagent.

Response: The connecting linker between NBD1 and the R helix could be readily observed in the full map without sharpening at a reasonable contour level, as shown in the revised Supplementary Fig. 2a-b. As suggested by the reviewer, we have modeled the backbone of this linker in the pdb file. We ruled out the possibility that R helix is a contaminant for two reasons: first, we have never observed such density in SUR1 with the same purification protocol; second, R helix density is continuously connected to NBD1. We have included this discussion in lines 114-120 of the revised manuscript.

For the reviewer’s inspection, we have uploaded the revised pdb file and unsharpened full map to Dropbox via the following link:

see editor email for link to files

2. For the RMSD values, the authors seem to be confused a bit. I was talking about the RMSD between SUR2A_{IF}/MgATP/MgATP and SUR2A_{IF}/MgATP/MgADP, not the difference between NBD1 and NBD2. In addition, although the RMSD values are shown in the figure, there is no description in the text to explain it, which is odd.

Response: We have included the RMSD of both the whole molecule and the NBD layer at lines 153 and 158 in the revised manuscript. We think the reviewer was questioning whether the conformation of SUR2A_{IF}/MgATP/MgADP is the same as that of SUR2A_{IF}/MgATP/MgATP. To answer this question, we have provided the comparison of their electron density maps low-pass filtered to the same 6 Å resolution in the revised Supplementary Fig. 4k-n, showing the obvious structural differences in the NBD layer. We have also uploaded these two maps into the same Dropbox folder for the reviewer’s inspection.

3. I am not convinced about the structure of SUR2A_{IF}/MgATP/MgADP, especially the nucleotide-binding status. The authors “speculated” that the NBD2 is bound with MgADP because NBDs are not exactly the same as in SUR2A_{IF}/MgATP/MgATP (and the first class of SUR2A_{IF}/MgATP/MgADP).

Such structural differences could probably be simply because of the resolution difference between the classes. Without the figure shown (supp fig3g,h), it is safe to say that the density is too bad to fit anything (or you can fit any nucleotides you like).

Response: The same as the answer to question #2, we have provided the comparison of maps of $SUR2A_{IF/MgATP/MgADP}$ and $SUR2A_{IF/MgATP/MgATP}$ in the revised Supplementary Fig. 4k-n, showing the large differences in the NBD layer. Furthermore, we have also clearly described our logic for assigning the second class as $SUR2A_{IF/MgATP/MgADP}$ in lines 161-173 and warned the reader about the potential caveat of such an assignment: “Unfortunately, the local map quality of NBDs was not sufficient for the assignment of the nucleotide identities purely based on the electron density map (Supplementary Fig. 4g-h). Because NBD1 was always bound with Mg-ATP even in the presence of a high concentration of Mg-ADP³², we assigned the nucleotide bound at NBD1 as Mg-ATP. Moreover, because of the different conformation of the second 3D class compared with the first 3D class ($SUR2A_{IF/MgATP/MgATP}$), we reasoned that Mg-ADP rather than Mg-ATP should bind at NBD2, otherwise the second 3D class would have the same structure as $SUR2A_{IF/MgATP/MgATP}$. Notably, although the assignment of Mg-ADP at NBD2 is logically reasonable and the structural models, especially the coordinates of nucleotides, were refined against the cryo-EM maps to reasonable geometry, we suggest cautious interpretation of the nucleotide-binding poses, because of the large positional uncertainty intrinsic to this local map quality (Supplementary Fig. 4g-h). Based on the discussion aforementioned, we tentatively assign the second 3D class as $SUR2A_{IF/MgATP/MgADP}$.”

Overall, the rebuttal from the authors does not alleviate my concern about the “over-interpretation” of the data.

Response: We think the new maps, new figures, and new discussions provided in the revision further strengthened our conclusions.